# Germline variation at 8q24 and prostate cancer risk in men of European ancestry

Marco Matejcic, Edward J. Saunders et al.[#]

Chromosome 8q24 is a susceptibility locus for multiple cancers, including prostate cancer. Here we combine genetic data across the 8q24 susceptibility region from 71,535 prostate cancer cases and 52,935 controls of European ancestry to define the overall contribution of germline variation at 8q24 to prostate cancer risk. We identify 12 independent risk signals for prostate cancer ($p < 4.28 \times 10^{-15}$), including three risk variants that have yet to be reported. From a polygenic risk score (PRS) model, derived to assess the cumulative effect of risk variants at 8q24, men in the top 1% of the PRS have a 4-fold (95%CI = 3.62–4.40) greater risk compared to the population average. These 12 variants account for ~25% of what can be currently explained of the familial risk of prostate cancer by known genetic risk factors. These findings highlight the overwhelming contribution of germline variation at 8q24 on prostate cancer risk which has implications for population risk stratification.

Prostate cancer (PCa) is the most common cancer among men in the US, with 161,360 new cases and 26,730 related deaths estimated in 2017[1]. Familial and epidemiological studies have provided evidence of substantial heritability of PCa[2], and ~170 common risk loci have been identified through genome-wide association studies (GWAS)[3]. The susceptibility region on chromosome 8q24 has been shown to be a major contributor to PCa risk, with multiple variants clustered in five linkage disequilibrium (LD) blocks spanning ~600 Mb that are independently associated with risk[4]. Many of these association signals reported at 8q24 have been replicated across racial/ethnic populations[5,6], pointing to common shared functional variants within 8q24. However, rare ancestry-specific variants have also been detected, which confer larger relative risks of PCa (odds ratios [ORs] >2.0) than common risk variants in the region and signify allelic heterogeneity in the contribution of germline variation at 8q24 to PCa risk across populations[7].

In the current study, we perform a comprehensive investigation of genetic variation across the 1.4 Mb cancer susceptibility region at 8q24 (127.6–129.0 Mb) in relation to PCa risk. We combine genotyped and imputed data from two large GWAS consortia (PRACTICAL/ELLIPSE OncoArray and iCOGS) including >124,000 individuals of European ancestry to search for novel risk variants, as well as to determine the overall contribution of genetic variation at 8q24 to PCa heritability. Our findings underscore the sizable impact of genetic variation in the 8q24 region in explaining inter-individual differences in PCa risk, with potential clinical utility for genetic risk prediction.

## Results

**Marginal and conditional association analysis.** Genotype data from the Illumina OncoArray and iCOGS array and imputation to 1000 Genomes Project (1KGP) were generated among 71,535 PCa cases and 52,935 controls of European ancestry from 86 case-control studies (see Methods). Of the 5600 genotyped and imputed variants at 8q24 (127.6–129.0 Mb) with minor allele frequency (MAF) > 0.1% retained for analysis (see Methods), 1268 (23%) were associated with PCa risk at $p < 5 \times 10^{-8}$ while 2772 (49%) were marginally associated at $p < 0.05$. These 5600 markers capture, at $r^2 > 0.8$, 90% and 97% of all variants at 8q24

(127.6–129.0 Mb) with MAF ≥ 1% and ≥5%, respectively (based on 1KGP Phase 3 EUR panel). In a forward and backward stepwise selection model on variants marginally associated with PCa risk ($p < 0.05$, $n = 2772$; see Methods), we identified 12 variants with conditional $p$-values from the Wald test between $2.93 \times 10^{-137}$ and $4.28 \times 10^{-15}$ (Table 1). None of the other variants were statistically significant at $p < 5 \times 10^{-8}$ after adjustment for the 12 independent hits (Fig. 1). The 8q24 region is shown in Supplementary Fig. 1. Of these 12 stepwise signals, three had alleles with extreme risk allele frequencies (RAFs) that conveyed large effects (rs77541621, RAF = 2%, OR = 1.85, 95%CI = 1.76–1.94; rs183373024, RAF = 1%, OR = 2.67, 95%CI = 2.43–2.93; rs190257175, RAF = 99%, OR = 1.60, 95%CI = 1.42–1.80). The remaining variants had RAFs between 0.11 and 0.92 and conditional ORs that were more modest and ranged from 1.10 to 1.37 (Table 1). For 8 of the 12 variants, the allele found to be positively associated with PCa risk was the predominant allele (i.e., >50% in frequency). For two variants, rs78511380 and rs190257175, the marginal associations were not genome-wide significant and substantially weaker than those in the conditional model. For rs78511380, the marginal OR was slightly protective (OR = 0.97; $p = 0.027$), but reversed direction and was highly statistically significant when conditioning on the other 11 variants (OR = 1.19; $p = 3.5 \times 10^{-18}$; Table 1).

**Haplotype analysis.** The haplotype analysis showed an additive effect of the 12 independent risk variants consistent with that predicted in the single variant test; co-occurrence of the 8q24 risk alleles on the same haplotype does not further increase the risk of PCa (Supplementary Table 1). The unique haplotype carrying the reference allele for rs190257175 (GCTTAT, 0.5% frequency) is also the sole haplotype associated with a reduced risk of PCa, suggesting that having the C allele confers a protective effect. The reference allele for rs78511380 (A, 8% frequency) occurs on a haplotype in block 2 together with the risk alleles for rs190257175, rs72725879 and rs5013678 (haplotype GTTTAA, 8%) which obscures the positive association with the T allele of rs78511380. Thus, the marginal protective effect associated with the risk allele for rs78511380 reflects an increased risk associated with the occurrence on a risk haplotype with other risk alleles (Supplementary Table 1).

---

**Table 1 Marginal and conditional estimates for genetic markers at 8q24 independently associated with prostate cancer risk**

| Variant ID[a] | Position[b] | Allele[c] | RAF[d] | LD cluster[e] | Conditional association[f] | | Marginal association | |
|---|---|---|---|---|---|---|---|---|
| | | | | | OR (95%CI)[g] | *p*-value | OR (95%CI)[h] | *p*-value |
| rs1914295 | 127910317 | T/C | 0.68 | block 1 | 1.10 (1.08–1.12) | $7.30 \times 10^{-25}$ | 1.09 (1.07–1.11) | $3.07 \times 10^{-21}$ |
| rs1487240 | 128021752 | A/G | 0.74 | block 1 | 1.20 (1.17–1.22) | $2.77 \times 10^{-66}$ | 1.16 (1.14–1.18) | $2.97 \times 10^{-54}$ |
| rs77541621 | 128077146 | A/G | 0.02 | block 2 | 1.85 (1.76–1.94) | $2.93 \times 10^{-137}$ | 1.83 (1.74–1.92) | $4.33 \times 10^{-137}$ |
| rs190257175 | 128103466 | T/C | 0.99 | block 2 | 1.60 (1.42–1.80) | $4.28 \times 10^{-15}$ | 1.36 (1.22–1.53) | $6.90 \times 10^{-8}$ |
| rs72725879 | 128103969 | T/C | 0.18 | block 2 | 1.31 (1.28–1.35) | $1.26 \times 10^{-83}$ | 1.17 (1.14–1.19) | $3.96 \times 10^{-48}$ |
| rs5013678 | 128103979 | T/C | 0.78 | block 2 | 1.10 (1.08–1.13) | $1.58 \times 10^{-19}$ | 1.20 (1.17–1.22) | $4.44 \times 10^{-68}$ |
| rs183373024 | 128104117 | G/A | 0.01 | block 2 | 2.67 (2.43–2.93) | $4.89 \times 10^{-95}$ | 3.20 (2.92–3.50) | $6.60 \times 10^{-138}$ |
| rs78511380 | 128114146 | T/A | 0.92 | block 2 | 1.19 (1.14–1.23) | $3.48 \times 10^{-18}$ | 0.97 (0.94–1.00) | 0.027 |
| rs17464492 | 128342866 | A/G | 0.72 | block 3 | 1.16 (1.14–1.18) | $3.01 \times 10^{-52}$ | 1.17 (1.15–1.19) | $9.05 \times 10^{-61}$ |
| rs6983267 | 128413305 | G/T | 0.51 | block 4 | 1.18 (1.16–1.20) | $5.68 \times 10^{-84}$ | 1.23 (1.21–1.25) | $3.15 \times 10^{-135}$ |
| rs7812894 | 128520479 | A/T | 0.11 | block 5 | 1.37 (1.33–1.40) | $1.55 \times 10^{-122}$ | 1.45 (1.41–1.49) | $1.20 \times 10^{-181}$ |
| rs12549761 | 128540776 | C/G | 0.87 | block 5 | 1.21 (1.18–1.24) | $1.61 \times 10^{-45}$ | 1.28 (1.25–1.31) | $1.38 \times 10^{-78}$ |

[a]Variants that remained genome-wide significantly associated with PCa risk ($p < 10^{-8}$) in the final stepwise model
[b]Chromosome position based on human genome build 37
[c]Risk allele/reference allele
[d]Risk allele frequency
[e]LD clusters were inferred based on recombination hotspots using Haploview 4.2[29] and defined as previously reported by Al Olama et al.[4]
[f]Each variant was incorporated in the stepwise model based on the strength of marginal association from the meta-analysis of OncoArray and iCOGS data
[g]Per-allele odds ratio and 95% confidence interval adjusted for country, 7(OncoArray)/8(iCOGS) principal components and all other variants in the table
[h]Per-allele odds ratio and 95% confidence interval adjusted for country and 7(OncoArray)/8(iCOGS) principal components

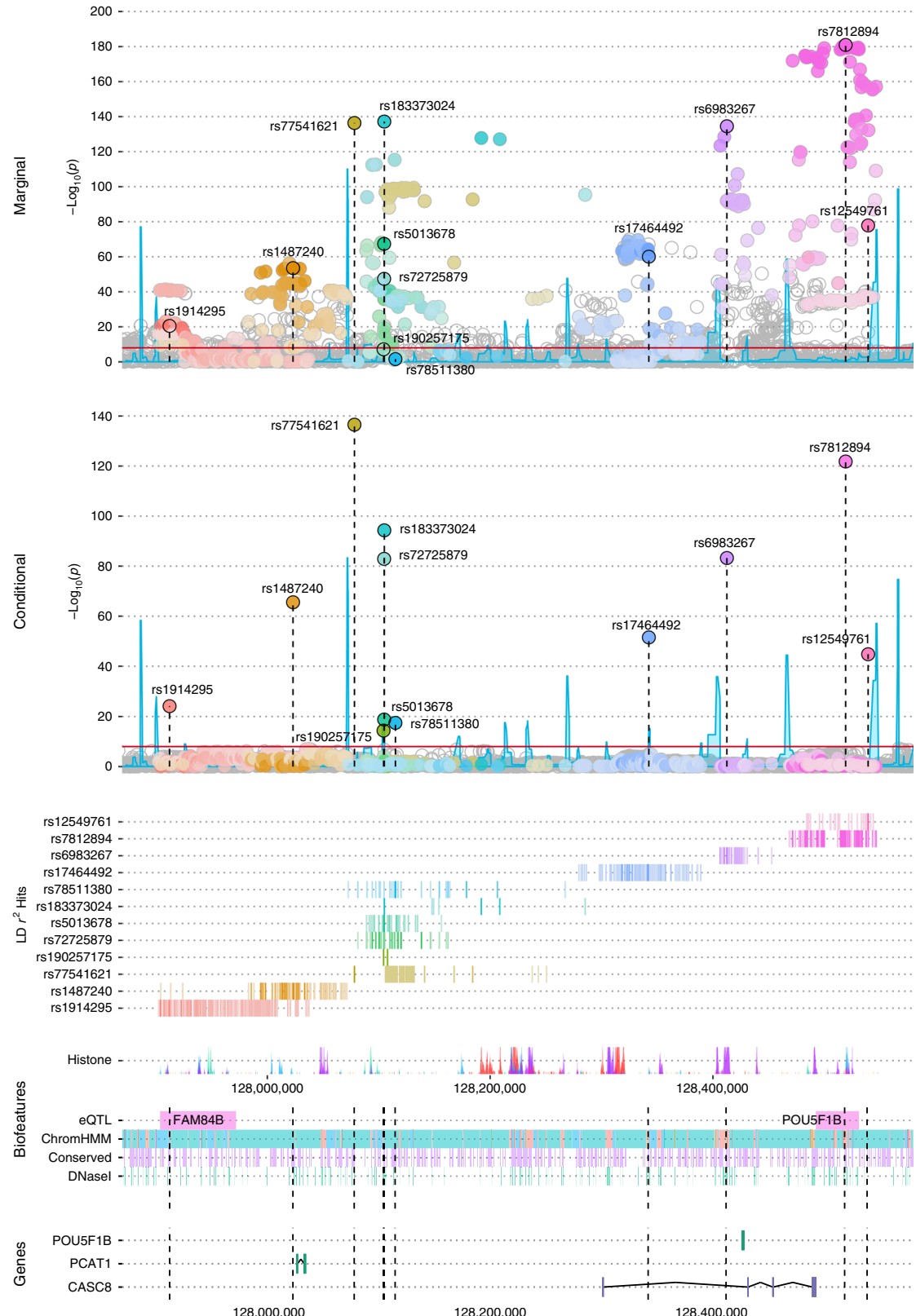

**Correlation with known risk loci**. The 12 risk variants spanned across the five LD blocks previously reported to harbor risk variants for PCa at 8q24[4], with block 2 harboring six signals, blocks 1 and 5 two signals each, and blocks 3 and 4 only one (Supplementary Fig. 2). Except for a weak correlation between rs72725879 and rs78511380 in block 2 ($r^2 = 0.28$), the risk variants were uncorrelated with each other ($r^2 \leq 0.09$; Supplementary Data 1), which corroborates their independent association with PCa risk. Eight of the variants (rs1487240, rs77541621, rs72725879, rs5013678, rs183373024, rs17464492, rs6983267,

**Fig. 1** LocusExplorer plots of the 12 variants at 8q24 significantly associated with PCa risk. 'Marginal' and 'Conditional' Manhattan plot panels show marginal and conditional association results, respectively. Variant positions (x-axis) and $-\log_{10}$ p-values from the Wald test (y-axis) are shown, with the red line indicating the threshold for genome-wide significant association with PCa risk ($p \leq 5 \times 10^{-8}$) and blue peaks local estimates of recombination rates. The position of the 12 independent variants is labeled in each plot. Clusters of correlated variants for each independent signal are distinguished using different colors and also depicted on the 'LD $r^2$ Hits' track. Stronger shading indicates greater correlation with the lead variant, with variants not correlated at $r^2 \geq 0.2$ with any lead variant uncolored. Pairwise correlations are based on the European ancestry (EUR) panel from the 1000 Genomes Project (1KGP) Phase 3. The relative position of RefSeq genes and biological annotations are shown in the 'Genes' and 'Biofeatures' panels, respectively. Genes on the positive strand are denoted in green and those on the negative strand in purple. Annotations displayed are: histone modifications in ENCODE tier 1 cell lines (Histone track), the positions of any variants that were eQTLs with prostate tumor expression in TCGA prostate adenocarcinoma samples and the respective genes for which expression is altered (eQTL track), chromatin state categorizations in the PrEC cell-line by ChromHMM (ChromHMM track), the position of conserved element peaks (Conserved track) and the position of DNaseI hypersensitivity site peaks in ENCODE prostate cell-lines (DNaseI track). The data displayed in this plot may be explored interactively through the LocusExplorer application (http://www.oncogenetics.icr.ac.uk/8q24/)

---

### Table 2 Relative risk of PCa for polygenic risk score (PRS) groups

| Risk category percentile[a] | No. of individuals | | Risk estimates for PRS groups | |
|---|---|---|---|---|
| | Controls | Cases | OR (95% CI)[b] | *p*-value |
| ≤1% | 530 | 339 | 0.52 (0.45–0.59) | $2.11 \times 10^{-20}$ |
| 1%–10% | 4771 | 3636 | 0.62 (0.59–0.65) | $6.26 \times 10^{-90}$ |
| 10%–25% | 7936 | 7359 | 0.75 (0.72–0.78) | $3.62 \times 10^{-54}$ |
| 25%–75% | 26,464 | 32,743 | 1.00 (Ref) | |
| 75%–90% | 7940 | 13,431 | 1.37 (1.32–1.41) | $6.55 \times 10^{-77}$ |
| 90%–99% | 4766 | 11,451 | 1.93 (1.86–2.01) | $4.13 \times 10^{-249}$ |
| >99% | 528 | 2576 | 3.99 (3.62–4.40) | $5.64 \times 10^{-172}$ |

*Note*: PRS were calculated for variants from the final stepwise model with allele dosage from OncoArray and iCOGS weighted by the per-allele conditionally adjusted odds ratios from the meta-analysis
[a]Risk category groups were based on the percentile distribution of risk alleles in overall controls
[b]Estimated effect of each PRS group relative to the interquartile range (25–75%) in OncoArray and iCOGS datasets separately, and then meta-analyzed across the two studies; odds ratios were adjusted for country and 7(OncoArray)/8(iCOGS) principal components

---

### Table 3 Proportion of familial relative risk (FRR) and heritability ($h_g^2$) of PCa explained by known risk variants

| Source | No. of variants | Proportion of FRR (95%CI) | % of total FRR | $h_g^2$ (SE) | % of total $h_g^2$ |
|---|---|---|---|---|---|
| 8q24[a] | 12 | 9.42 (8.22–10.88) | 25.4 | 0.027 (0.011) | 22.2 |
| HOXB13[b] | 1 | 1.91 (1.20–2.85) | 5.2 | 0.004 (0.005) | 3.0 |
| All other variants[b,c] | 162 | 25.77 (22.94–29.36) | 69.5 | 0.092 (0.010) | 74.9 |
| Total | 175 | 37.08 (32.89–42.49) | 100 | 0.118 (0.012) | 100 |

[a]Conditional estimates were derived by fitting a single model with all variants from OncoArray data
[b]Risk estimates and allele frequencies for regions with a single variant are from a meta-analysis of OncoArray, iCOGS and 6 additional GWAS[3]
[c]Risk variants included from fine-mapping of PCa susceptibility loci in European ancestry populations[11]

---

rs7812894) have been previously reported either directly (Supplementary Table 2) or are correlated ($r^2 \geq 0.42$) with known markers of PCa risk from studies in populations of European, African or Asian ancestry (Supplementary Data 1)[4,7–10]. The marginal estimates for previously published PCa risk variants at 8q24 in the current study are shown in Supplementary Table 2. The variant rs1914295 in block 1 is only weakly correlated with the previously reported risk variants rs12543663 and rs10086908 ($r^2 = 0.17$ and 0.14, respectively), while rs7851380 is modestly correlated with the previously reported risk variant rs1016343 ($r^2 = 0.28$). The remaining two variants, rs190257175 and rs12549761, are not correlated ($r^2 < 0.027$) with any known PCa risk marker.

**Polygenic risk score and familial relative risk**. To estimate the cumulative effect of germline variation at 8q24 on PCa risk, a polygenic risk score (PRS) was calculated for the 12 independent risk alleles from the final model based on allele dosages weighted by the per-allele conditionally adjusted ORs (see Methods). Compared to the men at 'average risk' (i.e., the 25th–75th PRS range among controls), men in the top 10% of the PRS

distribution had a 1.93-fold relative risk (95%CI = 1.86–2.01) (Table 2), with the risk being 3.99-fold higher (95%CI = 3.62–4.40) for men in the top 1%. Risk estimates by PRS category were not modified by family history (FamHist-yes: OR = 4.24, 95%CI = 2.85–6.31; FamHist-no: OR = 3.38, 95%CI = 2.88–3.97). To quantify the impact of germline variation at 8q24, we also estimated the proportion of familial relative risk (FRR) and heritability of PCa contributed by 8q24 and compared this to the proportions explained by all known PCa risk variants including 8q24 (see Methods). The 175 established PCa susceptibility loci identified to date[3,11] are estimated to explain 37.08% (95%CI = 32.89–42.49) of the FRR of PCa, while the 12 independent signals at 8q24 alone capture 9.42% (95%CI = 8.22–10.88), which is 25.4% of the total FRR explained by known genetic risk factors for PCa (Table 3). This is similar to the proportion of heritability explained by 8q24 variants (22.2%) compared to the total explained heritability by the known risk variants (0.118). In comparison, the next highest contribution of an individual susceptibility region to the FRR of PCa is the *TERT* region at chromosome 5p15, where 5 independent signals contributed 2.63% (95%CI = 2.34–3.00). No other individual GWAS

locus has been established as explaining >2% of the FRR, including the low frequency, non-synonymous, moderate penetrance *HOXB13* variant (rs138213197) at chromosome 17q21 that is estimated to explain only 1.91% (95%CI = 1.20–2.85) of the FRR[11].

**JAM analysis.** We explored our data with a second fine-mapping approach, JAM (Joint Analysis of Marginal summary statistics)[12], which uses GWAS summary statistics to identify credible sets of variants that define the independent association signals in susceptibility regions (see Methods). The 95% credible set for the JAM analysis confirmed all of the independent signals from stepwise analysis except rs190257175, for which evidence for an association was weak (variant-specific Bayes factor (BF) = 1.17). There were 50 total variants included in the 95% credible set, and 174 after including variants in high LD ($r^2 > 0.9$) with those in the credible set (Supplementary Data 2).

## Discussion

In this large study of germline genetic variation across the 8q24 region, we identified 12 independent association signals among men of European ancestry, with three of the risk variants (rs1914295, rs190257175, and rs12549761) being weakly correlated ($r^2 \leq 0.17$) with known PCa risk markers. The combination of these 12 independent signals at 8q24 capture approximately one quarter of the total PCa FRR explained by known genetic risk factors, which is substantially greater than any other known PCa risk locus.

The 8q24 region is the major susceptibility region for PCa; however, the underlying biological mechanism(s) through which germline variation in this region influences PCa risk remains uncertain. For each of the 12 risk variants at 8q24, the 95% credible set defined noteworthy (i.e., putative functional) variants based on summary statistics while accounting for LD. To inform biological functionality, we overlaid epigenetic functional annotation using publicly available datasets (see Methods) with the location of the 12 independent signals (and corresponding 174 variants within their 95% credible sets; Supplementary Data 3). Of the 12 independent lead variants, 6 are situated within putative transcriptional enhancers in prostate cell-lines; either through intersection with H3K27AC (rs72725879, rs5013678, rs78511380, rs6983267 and rs7812894) or through a ChromHMM enhancer annotation (rs17464492, rs6983267, rs7812894). Eight of the 12 stepwise hits (rs77541621, rs190257175, rs5013678, rs183373024, rs78511380, rs17464492, rs6983267, rs7812894) also intersect transcription factor binding site peaks from multiple ChIP-seq datasets representing the AR, ERG, FOXA1, GABPA, GATA2, HOXB13, and NKX3.1 transcription factors, with all 8 intersecting a FOXA1 mark and half an AR binding site. These variants may therefore exert their effect through regulation of enhancer activity and long-range expression of genes important for cancer tumorigenesis and/or progression[13]. The variant rs6983267 has also been shown to act in an allele-specific manner to regulate prostate enhancer activity and expression of the proto-oncogene *MYC* in vitro and in vivo[14,15]. However, despite the close proximity to the *MYC* locus, no direct association has been detected between 8q24 risk alleles and *MYC* expression in normal and tumor human prostate tissues[16]. The rare variant with the largest effect on risk, rs183373024, shows high evidence of functionality based on overlap with multiple DNaseI and transcription factor binding site peaks (for AR, FOXA1, HOXB13, and NKX3.1), which supports previous findings of an allele-dependent effect of this variant on the disruption of a FOXA1 binding motif[17]. Seven independent signals (rs1914295, rs1487240, rs77541621, rs72725879, rs5013678, rs183373024,

rs78511380) and variants correlated at $r^2 > 0.9$ with these signals (Supplementary Data 2) are located within or near a number of prostate cancer–associated long noncoding RNAs (lncRNAs), including *PRNCR1*, *PCAT1*, and *CCAT2*, previously reported to be upregulated in human PCa cells[18] and tissues[19,20]. Based on eQTL annotations in prostate adenocarcinoma cells, the independent signal rs1914295 and three correlated variants ($r^2 > 0.9$; Supplementary Data 2) are associated with overexpression of *FAM84B*, a gene previously associated with progression and poor prognosis of PCa in animal studies[21]. Variants correlated at $r^2 > 0.9$ with rs7812894 ($n = 9$; Supplemental Table 4) are eQTLs for *POU5F1B*, a gene overexpressed in cancer cell lines and cancer tissues[22,23], although its role in PCa development is unknown. Whilst we have successfully refined the 8q24 region and identified a subset of variants with putative biological function within our credible set, multi-ethnic comparisons may help refine the association signals even further and precisely identify the functional alleles and biological mechanisms that modify PCa risk.

Whereas the individual associations of the 8q24 variants with PCa risk are relatively modest (ORs < 2.0, except for rs183373024), their cumulative effects are substantial, with risk being 4-fold higher for men in the top 1% of the 8q24-only PRS. The contribution to the overall FRR of PCa is substantially greater for the 8q24 region (9.42%) than for any other known GWAS locus, including the moderate penetrance non-synonymous variant in *HOXB13* (1.91%). The ability of these markers to explain ~25.4% of what can be currently explained by all known PCa risk variants is a clear indication of the important contribution of germline variation at 8q24 on PCa risk. Our study was predominantly powered to analyze variants with MAF > 1% as the imputed variants with MAF = 0.1-1% were most likely to fail quality control (QC); however, the high density of genotyped markers and haplotypes at 8q24 in the OncoArray and iCOGS studies provided a robust backbone for imputation and increased the chances to impute lower MAF variants with high imputation quality score. Understanding of the biology of these variants and the underlying genetic basis of PCa could provide new insights into the identification of reliable risk-prediction biomarkers for PCa, as well as enable the development of effective strategies for targeted screening and prevention.

## Methods

**Study subjects, genotyping, and quality control.** We combined genotype data from the PRACTICAL/ELLIPSE OncoArray and iCOGS consortia[3,24], which included 143,699 men of European ancestry from 86 case-control studies largely based in either the US or Europe. In each study, cases primarily included men with incident PCa while controls were men without a prior diagnosis of the disease.

Both of the OncoArray and iCOGS custom arrays were designed to provide high coverage of common alleles (minor allele frequency [MAF] > 5%) across 8q24 (127.6–129.0 Mb) based on the 1000 Genomes Project (1KGP) Phase 3 for OncoArray, and the European ancestry (EUR) panel from HapMap Phase 2 for iCOGS. A total of 57,580 PCa cases and 37,927 controls of European ancestry were genotyped with the Illumina OncoArray, and 24,198 PCa cases and 23,994 controls of European ancestry were genotyped with the Illumina iCOGS array. For both studies, sample exclusion criteria included duplicate samples, first-degree relatives, samples with a call rate <95% or with extreme heterozygosity ($p < 10^{-6}$), and samples with an estimated proportion of European ancestry <0.8[3,24]. In total, genotype data for 53,449 PCa cases and 36,224 controls from OncoArray and 18,086 PCa cases and 16,711 controls from iCOGS were included in the analysis. Genetic variants with call rates <0.95, deviation from Hardy-Weinberg equilibrium ($p < 10^{-7}$ in controls), and genotype discrepancy in >2% of duplicate samples were excluded. Of the final 498,417 genotyped variants on the OncoArray and 201,598 on the iCOGS array that passed QC, 1581 and 1737 within the 8q24 region, respectively, were retained for imputation.

All studies complied with all relevant ethical regulations and were approved by the institutional review boards at each of the participating institutions. Informed consent was obtained from all study participants. Additional details of each study are provided in the Supplementary Note 1.

**Imputation analysis.** Imputation of both OncoArray and iCOGS genotype data was performed using SHAPEIT[25] and IMPUTEv2[26] to the October 2014 (Phase 3)

release of the 1KGP reference panel. A total of 10,136 variants from OncoArray and 10,360 variants from iCOGS with MAF > 0.1% were imputed across the risk region at 8q24 (127.6–129.0 Mb). Variants with an imputation quality score >0.8 were retained for a total of 5600 overlapping variants between the two datasets.

**Statistical analysis**. Unconditional logistic regression was used to estimate per-allele odds ratios (ORs) and 95% confidence intervals (CIs) for the association between genetic variants (single nucleotide polymorphisms and insertion/deletion polymorphisms) and PCa risk adjusting for country and principal components (7 for OncoArray and 8 for iCOGS). Allele dosage effects were tested through a 1-degree of freedom two-tailed Wald trend test. The marginal risk estimates for the 5600 variants at 8q24 that passed QC were combined by a fixed effect meta-analysis with inverse variance weighting using METAL[27]. A modified forward and backward stepwise model selection with inclusion and exclusion criteria of $p \leq 5 \times 10^{-8}$ was performed on variants marginally associated with PCa risk from the meta results ($p < 0.05$, $n = 2772$). At each step, the effect estimates for the candidate variants from both studies (OncoArray and iCOGS) were meta-analyzed and each variant was incorporated into the model based on the strength of association. All remaining variants were included one-at-a-time into the logistic regression model conditioning on those already incorporated in the model. We applied a conservative threshold for independent associations, with variants kept in the model if their meta p-value from the Wald test was genome-wide significant at $p \leq 5 \times 10^{-8}$ after adjustment for the other variants in the model. Correlations between variants in the final model and previously published PCa risk variants at 8q24 were estimated using the 1KGP Phase 3 EUR panel (Supplementary Data 1).

**Haplotype analysis**. Haplotypes were estimated in the Oncoarray data only using variants from the final stepwise model selection ($n = 12$) and the EM algorithm[28] within LD block regions inferred based on recombination hotspots using Haploview 4.2 (Broad Institute, Cambridge, MA, USA)[29]. Only haplotypes with an estimated frequency ≥0.5% were tested.

**Polygenic risk score and familial relative risk**. An 8q24-only polygenic risk score (PRS) was calculated for variants from the final model ($n = 12$) with allele dosage from OncoArray and iCOGS weighted by the per-allele conditionally adjusted ORs from the meta-analysis. Categorization of the PRS was based on the percentile distribution in controls, and the risk for each category was estimated relative to the interquartile range (25–75%) in OncoArray and iCOGS separately, and then meta-analyzed across the two studies. We estimated the contribution of 8q24 variants to the familial (first-degree) relative risk (FRR) of PCa (FRR = 2.5)[30] under a multiplicative model, and compared this to the FRR explained by all known PCa risk variants including 8q24 (Supplementary Data 4). We also estimated heritability of PCa using the LMM approach as implemented in GCTA[31]. For regions which have been fine-mapped using the OncoArray meta-analysis data, we used the updated representative lead variants, otherwise the originally reported variant was included provided that it had replicated at genome-wide significance in the meta-analysis; this yielded a total of 175 independently associated PCa variants for the FRR and heritability calculations[3,11]. For these analyses, we used conditional estimates from fitting a single model with all variants in the OncoArray dataset for regions with multiple variants and the overall marginal meta-analysis results from Schumacher et al.[3] for regions with a single variant. To correct for potential bias in effect estimation of newly discovered variants, we implemented a Bayesian version of the weighted correction[32], which incorporates the uncertainty in the effect estimate into the final estimates of the bias-corrected ORs, 95%CIs and the corresponding calculations of percent FRR explained.

**JAM analysis**. To confirm the stepwise results and identify candidate variants for potential functional follow-up, we used a second fine-mapping approach, JAM (Joint Analysis of Marginal summary statistics)[12]. JAM is a multivariate Bayesian variable selection framework that uses GWAS summary statistics to identify the most likely number of independent associations within a locus and define credible sets of variants driving those associations. JAM was applied to summary statistics from the meta-analysis results using LD estimated from imputed individual level data from 20,000 cases and 20,000 controls randomly selected from the OncoArray sub-study. LD pruning was performed using Priority Pruner (http://prioritypruner.sourceforge.net/) on the 2772 marginally associated variants at $r^2 = 0.9$, resulting in 825 tag variants analyzed in four independent JAM runs with varying starting seeds. Credible sets were determined as the tag variants that were selected in the top models that summed to a specific cumulative posterior probability in all four of the independent JAM runs, plus their designated high LD proxy variants from the pruning step.

**Functional annotation**. Variants in the 95% credible set ($n = 50$) plus variants correlated at $r^2 > 0.9$ with those in the credible set ($n = 174$) were annotated for putative evidence of biological functionality using publicly available datasets as described by Dadaev et al.[11]. Briefly, variants were annotated for proximity to gene (GENCODEv19), miRNA transcripts (miRBase release 20), evolutionary constraint (according to GERP++, SiPhy and PhastCons algorithms), likelihood of

pathogenicity (CADDv1.3) and overlap with prospective regulatory elements in prostate-specific datasets (DNaseI hypersensitivity sites, H3K27Ac, H3K27me3 and H3K4me3 histone modifications, and for AR, CTCF, ERG, FOXA1, GABPA, GATA2, HOXB13, and NKX3.1 transcription factor binding sites) in a mixture of LNCaP, PC-3, PrEC, RWPE1, and VCaP cell lines and human prostate tumor tissues downloaded from the Cistrome Data Browser (http://cistrome.org/db/). The chromatin state in which each variant resides was assessed using ChromHMM annotations from two prostate cell lines (PrEC and PC3). Cis-gene regulation was evaluated using 359 prostate adenoma cases from The Cancer Genome Atlas (TCGA PRAD; https://gdc-portal.nci.nih.gov) that passed QC[11]. The eQTL analysis was performed using FastQTL with 1000 permutations for each gene within a 1Mb window. We then used the method by Nica et al.[33] that integrates eQTLs and GWAS results in order to reveal the subset of association signals that are due to cis eQTLs. For each significant eQTL, we added the candidate variant to the linear regression model to assess if the inclusion better explains the change in expression of the gene. We retrieved the p-value of the model, assigning p-value of 1 if the eQTL and variant are the same. Then we ranked the p-values in descending order for each eQTL, and finally calculated the colocalization score for each pair of eQTL and variants. In general, if an eQTL and candidate variant represent the same signal, this will be reflected by the variant having a high p-value, a low rank and consequently a high colocalization score.

## Data availability
The authors declare that data supporting the findings of this study are available within the paper [and in the supplementary information files]. However, some of the data used to generate the results of this study are available from the first author and the PRAC-TICAL Consortium upon request.

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

## Acknowledgements

Genotyping of the OncoArray was funded by the US National Institutes of Health (NIH) [U19 CA 148537 for ELucidating Loci Involved in Prostate Cancer SuscEptibility (ELLIPSE) project and X01HG007492 to the Center for Inherited Disease Research (CIDR) under contract number HHSN268201200008I]. Additional analytic support was provided by NIH NCI U01 CA188392 (PI: Schumacher). The PRACTICAL consortium (http://practical.icr.ac.uk/) was supported by Cancer Research UK Grants C5047/A7357, C1287/A10118, C1287/A16563, C5047/A3354, C5047/A10692, C16913/A6135, European Commission's Seventh Framework Programme grant agreement n° 223175 (HEALTH-F2-2009-223175), and The National Institute of Health (NIH) Cancer Post-Cancer GWAS initiative grant: No. 1 U19 CA 148537-01 (the GAME-ON initiative). We wish to thank all GWAS study groups contributing to the data set from which this study was conducted: OncoArray; iCOGS; The PRACTICAL (Prostate Cancer Association Group to Investigate Cancer-Associated Alterations in the Genome) Consortium; and The GAME-ON/ELLIPSE Consortium. Detailed acknowledgements and funding information for all GWAS study groups and from all the individual studies involved in the PRACTICAL Consortium are included in Supplementary Note 1. We would also like to thank the following for funding support: The Institute of Cancer Research and The Everyman Campaign, The Prostate Cancer Research Foundation, Prostate Research Campaign UK (now Prostate Action), The Orchid Cancer Appeal, The National Cancer Research Network UK, The National Cancer Research Institute (NCRI) UK. We are grateful for support of NIHR funding to the NIHR Biomedical Research Centre at The Institute of Cancer Research and The Royal Marsden NHS Foundation Trust.

## Author contributions

M.M. and E.J.S. contributed equally to this work. R.A.E., Z.K.-J., D.V.C., and C.A.H. jointly supervised this work. T.D. contributed with JAM analysis. M.N.B. contributed with FRR analysis. K.W. contributed with forward and backward stepwise selection. X.S. contributed with coverage analysis. A.A.A.O., F.R.S., S.A.I., K.G., S.B., S.I.B., D.A., S.K., K.M., V.L.S., S.M.G., C.M.T., J.B., J.C., H.G., N.P., J.S., A.W., C.W., L.Mu., P.K., G.C.-T., K.D.S., L.Ma., E.M.G., S.S.S., D.E.N., F.C.H., J.L.D., R.C.T., R.J.H., B.R., Y.-J.L., G.G.G., A.S.K., A.V., J.T.B., M.K., K.L.P., J.Y.P., J.L.S., C.C., B.G.N., H.B., C.M., J.K., M.R.T., S.L.N., K.D.R., A.R., L.F.N., D.L., R.K., N.U., F.C., P.A.T., M.G.D., M.J.R., F.M., K.-T.K., L.A.C.-A., H.P., S.N.T., D.J.S., The PRACTICAL Consortium, F.W., S.J.C., and D.F.E. were involved in sample and data collection.

## Additional information

Marco Matejcic[1], Edward J. Saunders[2], Tokhir Dadaev[2], Mark N. Brook[2], Kan Wang[1], Xin Sheng[1], Ali Amin Al Olama[3,4], Fredrick R. Schumacher[5,6], Sue A. Ingles[1], Koveela Govindasami[2], Sara Benlloch[2,3], Sonja I. Berndt[7], Demetrius Albanes[7], Stella Koutros[7], Kenneth Muir[8,9], Victoria L. Stevens[10], Susan M. Gapstur[10], Catherine M. Tangen[11], Jyotsna Batra[12,13], Judith Clements[12,13], Henrik Gronberg[14], Nora Pashayan[15,16], Johanna Schleutker[17,18,19], Alicja Wolk[20], Catharine West[21], Lorelei Mucci[22], Peter Kraft[23], Géraldine Cancel-Tassin[24,25], Karina D. Sorensen[26,27], Lovise Maehle[28], Eli M. Grindedal[28], Sara S. Strom[29], David E. Neal[30,31], Freddie C. Hamdy[32], Jenny L. Donovan[33], Ruth C. Travis[34], Robert J. Hamilton[35], Barry Rosenstein[36,37], Yong-Jie Lu[38], Graham G. Giles[39,40], Adam S. Kibel[41], Ana Vega[42], Jeanette T. Bensen[43], Manolis Kogevinas[44,45,46,47], Kathryn L. Penney[48], Jong Y. Park[49], Janet L. Stanford[50,51], Cezary Cybulski[52], Børge G. Nordestgaard[53,54], Hermann Brenner[55,56,57], Christiane Maier[58], Jeri Kim[59],

Manuel R. Teixeira[60,61], Susan L. Neuhausen[62], Kim De Ruyck[63], Azad Razack[64], Lisa F. Newcomb[50,65], Davor Lessel [66], Radka Kaneva[67], Nawaid Usmani[68,69], Frank Claessens[70], Paul A. Townsend[71], Manuela Gago-Dominguez[72,73], Monique J. Roobol [74], Florence Menegaux[75], Kay-Tee Khaw[76], Lisa A. Cannon-Albright[77,78], Hardev Pandha[79], Stephen N. Thibodeau[80], Daniel J. Schaid[81], The PRACTICAL Consortium, Fredrik Wiklund[14], Stephen J. Chanock [7], Douglas F. Easton [3,15], Rosalind A. Eeles [2,82], Zsofia Kote-Jarai[2], David V. Conti[1] & Christopher A. Haiman[1]

[1]Department of Preventive Medicine, Keck School of Medicine, University of Southern California/Norris Comprehensive Cancer Center, Los Angeles, CA 90033, USA. [2]The Institute of Cancer Research, London SW7 3RP, UK. [3]Centre for Cancer Genetic Epidemiology, Department of Public Health and Primary Care, Strangeways Research Laboratory, University of Cambridge, Cambridge CB1 8RN, UK. [4]Department of Clinical Neurosciences, University of Cambridge, Cambridge CB2 0QQ, UK. [5]Department of Population and Quantitative Health Sciences, Case Western Reserve University, Cleveland, OH 44106-7219, USA. [6]Seidman Cancer Center, University Hospitals, Cleveland, OH 44106, USA. [7]Division of Cancer Epidemiology and Genetics, National Cancer Institute, NIH, Bethesda, MD 20892, USA. [8]Institute of Population Health, University of Manchester, Manchester M13 9PL, UK. [9]Warwick Medical School, University of Warwick, Coventry CV4 7AL, UK. [10]Epidemiology Research Program, American Cancer Society, 250 Williams Street, Atlanta, GA 30303, USA. [11]SWOG Statistical Center, Fred Hutchinson Cancer Research Center, Seattle, WA 98109, USA. [12]Australian Prostate Cancer Research Centre-Qld, Institute of Health and Biomedical Innovation and School of Biomedical Science, Queensland University of Technology, Brisbane, QLD 4059, Australia. [13]Translational Research Institute, Brisbane, QLD 4102, Australia. [14]Department of Medical Epidemiology and Biostatistics, Karolinska Institute, SE-171 77 Stockholm, Sweden. [15]Centre for Cancer Genetic Epidemiology, Department of Oncology, Strangeways Research Laboratory, University of Cambridge, Cambridge CB1 8RN, UK. [16]Department of Applied Health Research, University College London, London WC1E 7HB, UK. [17]Department of Medical Biochemistry and Genetics, Institute of Biomedicine, University of Turku, FI-20014 Turku, Finland. [18]Tyks Microbiology and Genetics, Department of Medical Genetics, Turku University Hospital, 20521 Turku, Finland. [19]BioMediTech, University of Tampere, 33520 Tampere, Finland. [20]Division of Nutritional Epidemiology, Institute of Environmental Medicine, Karolinska Institutet, SE-171 77 Stockholm, Sweden. [21]Division of Cancer Sciences, Manchester Academic Health Science Centre, Radiotherapy Related Research, Manchester NIHR Biomedical Research Centre, The Christie Hospital NHS Foundation Trust, University of Manchester, Manchester M13 9PL, UK. [22]Department of Epidemiology, Harvard School of Public Health, Boston, MA 02115, USA. [23]Program in Genetic Epidemiology and Statistical Genetics, Department of Epidemiology, Harvard T.H. Chan School of Public Health, Boston, MA 02115, USA. [24]GRC N°5 ONCOTYPE-URO, UPMC Univ Paris 06, Tenon Hospital, F-75020 Paris, France. [25]CeRePP, Tenon Hospital, F-75020 Paris, France. [26]Department of Molecular Medicine, Aarhus University Hospital, 8200 Aarhus N, Denmark. [27]Department of Clinical Medicine, Aarhus University, 8200 Aarhus N, Denmark. [28]Department of Medical Genetics, Oslo University Hospital, 0424 Oslo, Norway. [29]Department of Epidemiology, The University of Texas MD Anderson Cancer Center, Houston, TX 77030, USA. [30]Department of Oncology, Addenbrooke's Hospital, University of Cambridge, Cambridge CB2 0QQ, UK. [31]Cancer Research UK Cambridge Research Institute, Li Ka Shing Centre, Cambridge CB2 0RE, UK. [32]Nuffield Department of Surgical Sciences, University of Oxford, Oxford OX1 2JD, UK. [33]School of Social and Community Medicine, University of Bristol, Canynge Hall, 39 Whatley Road, Bristol BS8 2PS, UK. [34]Cancer Epidemiology, Nuffield Department of Population Health, University of Oxford, Oxford OX3 7LF, UK. [35]Department of Surgical Oncology, Princess Margaret Cancer Centre, Toronto, ON M5G 2M9, Canada. [36]Department of Radiation Oncology, Icahn School of Medicine at Mount Sinai, New York, NY 10029, USA. [37]Department of Genetics and Genomic Sciences, Icahn School of Medicine at Mount Sinai, New York, NY 10029-5674, USA. [38]Centre for Molecular Oncology, John Vane Science Centre, Barts Cancer Institute, Queen Mary University of London, London EC1M 6BQ, UK. [39]Cancer Epidemiology & Intelligence Division, Cancer Council Victoria, Melbourne, VIC 3004, Australia. [40]Centre for Epidemiology and Biostatistics, Melbourne School of Population and Global Health, The University of Melbourne, Melbourne, VIC 3010, Australia. [41]Division of Urologic Surgery, Brigham and Womens Hospital, Boston, MA 02115, USA. [42]Fundación Pública Galega de Medicina Xenómica-SERGAS, Grupo de Medicina Xenómica, CIBERER, IDIS, 15706 Santiago de Compostela, Spain. [43]Department of Epidemiology, Gillings School of Global Public Health, University of North Carolina, Columbia, SC 29208, USA. [44]Centre for Research in Environmental Epidemiology (CREAL), Barcelona Institute for Global Health (ISGlobal), 08003 Barcelona, Spain. [45]CIBER Epidemiología y Salud Pública (CIBERESP), 28029 Madrid, Spain. [46]IMIM (Hospital del Mar Research Institute), 08003 Barcelona, Spain. [47]Universitat Pompeu Fabra (UPF), 08002 Barcelona, Spain. [48]Channing Division of Network Medicine, Department of Medicine, Brigham and Women's Hospital/ Harvard Medical School, Boston, MA 02184, USA. [49]Department of Cancer Epidemiology, Moffitt Cancer Center, Tampa, FL 33612, USA. [50]Division of Public Health Sciences, Fred Hutchinson Cancer Research Center, Seattle, WA 98109-1024, USA. [51]Department of Epidemiology, School of Public Health, University of Washington, Seattle, WA 98195, USA. [52]International Hereditary Cancer Center, Department of Genetics and Pathology, Pomeranian Medical University, 70-115 Szczecin, Poland. [53]Faculty of Health and Medical Sciences, University of Copenhagen, 2200 Copenhagen, Denmark. [54]Department of Clinical Biochemistry, Herlev and Gentofte Hospital, Copenhagen University Hospital, Herlev, 2200 Copenhagen, Denmark. [55]Division of Clinical Epidemiology and Aging Research, German Cancer Research Center (DKFZ), D-69120 Heidelberg, Germany. [56]German Cancer Consortium (DKTK), German Cancer Research Center (DKFZ), D-69120 Heidelberg, Germany. [57]Division of Preventive Oncology, German Cancer Research Center (DKFZ) and National Center for Tumor Diseases (NCT), 69120 Heidelberg, Germany. [58]Institute for Human Genetics, University Hospital Ulm, 89075 Ulm, Germany. [59]Department of Genitourinary Medical Oncology, The University of Texas MD Anderson Cancer Center, Houston, TX 77030, USA. [60]Department of Genetics, Portuguese Oncology Institute of Porto, 4200-072 Porto, Portugal. [61]Biomedical Sciences Institute (ICBAS), University of Porto, 4050-313 Porto, Portugal. [62]Department of Population Sciences, Beckman Research Institute of the City of Hope, Duarte, CA 91010, USA. [63]Ghent University, Faculty of Medicine and Health Sciences, Basic Medical Sciences, B-9000 Gent, Belgium. [64]Department of Surgery, Faculty of Medicine, University of Malaya, 50603 Kuala Lumpur, Malaysia. [65]Department of Urology, University of Washington, Seattle, WA 98195, USA. [66]Institute of Human Genetics, University Medical Center Hamburg-Eppendorf, D-20246 Hamburg, Germany. [67]Molecular Medicine Center, Department of Medical Chemistry and Biochemistry, Medical University of Sofia, 1431 Sofia, Bulgaria. [68]Department of Oncology, Cross Cancer Institute, University of Alberta, Edmonton, AB T6G 1Z2, Canada. [69]Division of Radiation Oncology, Cross Cancer Institute, University of Alberta, Edmonton, AB T6G 1Z2, Canada. [70]Molecular Endocrinology Laboratory, Department of Cellular and Molecular Medicine, KU Leuven, BE-3000 Leuven, Belgium. [71]Manchester Cancer Research Centre, Faculty of Biology Medicine and Health, Manchester Academic Health Science Centre, NIHR Manchester Biomedical Research Centre, Health Innovation Manchester, University of Manchester, Manchester M13 9WL, UK. [72]Genomic Medicine Group, Galician Foundation of Genomic Medicine, Instituto de Investigacion Sanitaria de Santiago de Compostela (IDIS), Complejo Hospitalario Universitario de Santiago, Servicio Galego de Saúde, SERGAS,

15706 Santiago de Compostela, Spain. [73]Moores Cancer Center, University of California San Diego, La Jolla, CA 92037, USA. [74]Department of Urology, Erasmus University Medical Center, 3015 CE Rotterdam, The Netherlands. [75]Cancer and Environment Group, Center for Research in Epidemiology and Population Health (CESP), INSERM, University Paris-Sud, University Paris-Saclay, 94807 Villejuif Cédex, France. [76]Clinical Gerontology Unit, University of Cambridge, Cambridge CB2 2QQ, UK. [77]Division of Genetic Epidemiology, Department of Medicine, University of Utah School of Medicine, Salt Lake City, UT 84112, USA. [78]George E. Wahlen Department of Veterans Affairs Medical Center, Salt Lake City, UT 84148, USA. [79]The University of Surrey, Guildford, Surrey GU2 7XH, UK. [80]Department of Laboratory Medicine and Pathology, Mayo Clinic, Rochester, MN 55905, USA. [81]Division of Biomedical Statistics and Informatics, Mayo Clinic, Rochester, MN 55905, USA. [82]Royal Marsden NHS Foundation Trust, London SW3 6JJ, UK. These authors contributed equally: Marco Matejcic, Edward J. Saunders. These authors jointly supervised this work: Rosalind A. Eeles, Zsofia Kote-Jarai, David V. Conti, Christopher A. Haiman. A full list of consortium members appears at the end of the paper.

## The PRACTICAL (Prostate Cancer Association Group to Investigate Cancer-Associated Alterations in the Genome) Consortium

Brian E. Henderson[1], Mariana C. Stern[1], Alison Thwaites[2], Michelle Guy[2], Ian Whitmore[2], Angela Morgan[2], Cyril Fisher[2], Steve Hazel[2], Naomi Livni[2], Margaret Cook[3], Laura Fachal[3,42], Stephanie Weinstein[7], Laura E. Beane Freeman[7], Robert N. Hoover[7], Mitchell J. Machiela[7], Artitaya Lophatananon[8,9], Brian D. Carter[10], Phyllis Goodman[11], Leire Moya[12,13], Srilakshmi Srinivasan[12,13], Mary-Anne Kedda[12,13], Trina Yeadon[12,13], Allison Eckert[12,13], Martin Eklund[14], Carin Cavalli-Bjoerkman[14], Alison M. Dunning[15], Csilla Sipeky[17], Niclas Hakansson[20], Rebecca Elliott[21], Hardeep Ranu[22], Edward Giovannucci[22], Constance Turman[23], David J. Hunter[23], Olivier Cussenot[24,25], Torben Falck Orntoft[26,27], Athene Lane[33], Sarah J. Lewis[33], Michael Davis[33], Tim J. Key[34], Paul Brown[35], Girish S. Kulkarni[35], Alexandre R. Zlotta[35], Neil E. Fleshner[35], Antonio Finelli[35], Xueying Mao[38], Jacek Marzec[38], Robert J. MacInnis[39,40], Roger Milne[39,40], John L. Hopper[40], Miguel Aguado[42], Mariona Bustamante[44], Gemma Castaño-Vinyals[44,45,46,47], Esther Gracia-Lavedan[44,45,46,47], Lluís Cecchini[46], Meir Stampfer[48], Jing Ma[48], Thomas A. Sellers[49], Milan S. Geybels[49], Hyun Park[49], Babu Zachariah[49], Suzanne Kolb[50], Dominika Wokolorczyk[52], Jan Lubinski[52], Wojciech Kluzniak[52], Sune F. Nielsen[53,54], Maren Weisher[54], Katarina Cuk[55], Walther Vogel[58], Manuel Luedeke[58], Christopher J. Logothetis[59], Paula Paulo[60], Marta Cardoso[60], Sofia Maia[60], Maria P. Silva[60], Linda Steele[62], Yuan Chun Ding[62], Gert De Meerleer[63], Sofie De Langhe[63], Hubert Thierens[63], Jasmine Lim[64], Meng H. Tan[64], Aik T. Ong[64], Daniel W. Lin[50,65], Darina Kachakova[67], Atanaska Mitkova[67], Vanio Mitev[67], Matthew Parliament[68,69], Guido Jenster[74], Christopher Bangma[74], F.H. Schroder[74], Thérèse Truong[75], Yves Akoli Koudou[75], Agnieszka Michael[79], Andrzej Kierzek[79], Ami Karlsson[79], Michael Broms[79], Huihai Wu[79], Claire Aukim-Hastie[79], Lori Tillmans[80], Shaun Riska[80], Shannon K. McDonnell[81], David Dearnaley[2,82], Amanda Spurdle[83], Robert Gardiner[84,85], Vanessa Hayes[86], Lisa Butler[87], Renea Taylor[88], Melissa Papargiris[88], Pamela Saunders[89], Paula Kujala[90], Kirsi Talala[91], Kimmo Taari[92], Søren Bentzen[93], Belynda Hicks[94], Aurelie Vogt[94], Amy Hutchinson[95], Angela Cox[96], Anne George[97], Ants Toi[98], Andrew Evans[99], Theodorus H. van der Kwast[99], Takashi Imai[100], Shiro Saito[101], Shan-Chao Zhao[102], Guoping Ren[103], Yangling Zhang[103], Yongwei Yu[104], Yudong Wu[105], Ji Wu[106], Bo Zhou[107], John Pedersen[108], Ramón Lobato-Busto[109], José Manuel Ruiz-Dominguez[110], Lourdes Mengual[111,112], Antonio Alcaraz[113], Julio Pow-Sang[114], Kathleen Herkommer[115], Aleksandrina Vlahova[116], Tihomir Dikov[116], Svetlana Christova[116], Angel Carracedo[42,117,118], Brigitte Tretarre[119], Xavier Rebillard[120], Claire Mulot[121], Jan Adolfsson[122,123], Par Stattin[124,125], Jan-Erik Johansson[126], Richard M. Martin[33,127,128], Ian M. Thompson Jr.[129], Suzanne Chambers[130,131], Joanne Aitken[130,131], Lisa Horvath[132,133], Anne-Maree Haynes[86,133], Wayne Tilley[134], Gail Risbridger[135,136], Markus Aly[14,137], Tobias Nordström[14,138], Paul Pharoah[3,139], Teuvo L.J. Tammela[140], Teemu Murtola[140,141], Anssi Auvinen[142], Neil Burnet[143], Gill Barnett[143], Gerald Andriole[144], Aleksandra Klim[144], Bettina F. Drake[144], Michael Borre[27,145], Sarah Kerns[146], Harry Ostrer[147], Hong-Wei Zhang[148], Guangwen Cao[148], Ji Lin[148], Jin Ling[148], Meiling Li[148], Ninghan Feng[149], Jie Li[150], Weiyang He[150], Xin Guo[150,151], Zan Sun[151],

Guomin Wang[152], Jianming Guo[152], Melissa C. Southey[153], Liesel M. FitzGerald[40,154], Gemma Marsden[32,155], Antonio Gómez-Caamaño[156], Ana Carballo[156], Paula Peleteiro[156], Patricia Calvo[156], Robert Szulkin[157,158], Javier Llorca[45,159], Trinidad Dierssen-Sotos[45,159], Ines Gomez-Acebo[45,159], Hui-Yi Lin[160], Elaine A. Ostrander[161], Rasmus Bisbjerg[162], Peter Klarskov[162], Martin Andreas Røder[163], Peter Iversen[53,163], Bernd Holleczek[164], Christa Stegmaier[164], Thomas Schnoeller[165], Philipp Bohnert[165], Esther M. John[166,167], Piet Ost[168], Soo-Hwang Teo[169], Marija Gamulin[170], Tomislav Kulis[171], Zeljko Kastelan[171], Chavdar Slavov[172], Elenko Popov[172], Thomas Van den Broeck[70,173], Steven Joniau[173], Samantha Larkin[174], Jose Esteban Castelao[175], Maria Elena Martinez[176], Ron H.N. van Schaik[177], Jianfeng Xu[178], Sara Lindström[179], Elio Riboli[180], Clare Berry[180], Afshan Siddiq[181], Federico Canzian[182], Laurence N. Kolonel[183], Loic Le Marchand[183], Matthew Freedman[184], Sylvie Cenee[75,185] & Marie Sanchez[75,185]

[83]Molecular Cancer Epidemiology Laboratory, QIMR Berghofer Institute of Medical Research, Herston, QLD 4006, Australia. [84]School of Medicine, University of Queensland, Herston, QLD 4006, Australia. [85]Royal Brisbane and Women's Hospital, Herston, QLD 4029, Australia. [86]The Kinghorn Cancer Centre (TKCC), Victoria, NSW 2010, Australia. [87]Prostate Cancer Research Group, South Australian Health and Medical Research Institute, Adelaide, SA 5000, Australia. [88]Department of Physiology, Biomedicine Discovery Institute, Cancer Program, Monash University, Melbourne, VIC 3800, Australia. [89]University of Adelaide, North Terrace, Adelaide, SA 5005, Australia. [90]Fimlab Laboratories, Tampere University Hospital, FI-33520 Tampere, Finland. [91]Finnish Cancer Registry, FI-00130 Helsinki, Finland. [92]Department of Urology, Helsinki University Central Hospital and University of Helsinki, FI-00014 Helsinki, Finland. [93]Division of Biostatistics and Bioinformatics, University of Maryland Greenebaum Cancer Center, and Department of Epidemiology and Public Health, University of Maryland School of Medicine, Baltimore, MD 21201, USA. [94]Cancer Genomics Research Laboratory (CGR), Division of Cancer Epidemiology and Genetics, FNLCR Leidos Biomedical Research, National Cancer Institute, Frederick, MD 21701, USA. [95]DNA Extraction and Staging Laboratory (DESL), Cancer Genomics Research Laboratory (CGR), Division of Cancer Epidemiology and Genetics, FNLCR Leidos Biomedical Research, National Cancer Institute, Frederick, MD 21701, USA. [96]Sheffield Institute for Nucleic Acids, University of Sheffield, Sheffield S10 2TN, UK. [97]Cambridge Cancer Trials Centre, Cambridge Clinical Trials Unit-Cancer Theme, Cambridge University Hospitals NHS Foundation Trust, Cambridge CB2 0QQ, UK. [98]Department of Medical Imaging, University Health Network, Toronto, ON M5G 2C4, Canada. [99]Department of Pathology, University Health Network, Toronto, ON M5G 2C4, Canada. [100]Advanced Radiation Biology Research Program, Research Center for Charged Particle Therapy, National Institute of Radiological Sciences, Chiba 263-8555, Japan. [101]Department of Urology, National Hospital Organization Tokyo Medical Center, Tokyo 152-8902, Japan. [102]Department of Urology, Nanfang Hospital, Southern Medical University, 510515 Guangzhou, China. [103]Department of Pathology, The First Affiliated Hospital, Zhejiang University Medical College, 310009 Hangzhou, China. [104]Department of Pathology, Changhai Hospital, The Second Military Medical University, 200433 Shanghai, China. [105]Department of Urology, First Affiliated Hospital, Medical College, Zhengzhou University, 450003 Zhengzhou, China. [106]Department of Urology, North Sichuan Medical College, 637000 Nanchong, China. [107]Department of Nutrition Science, Shenyang Medical College, 110034 Shenyang, China. [108]Tissupath Pty Ltd., Melbourne, VIC 3122, Australia. [109]Department of Medical Physics, Complexo Hospitalario Universitario de Santiago, SERGAS, 15706 Santiago de Compostela, Spain. [110]Urology Department, Hospital Germans Trias I Pujol, 08916 Barcelona, Spain. [111]Laboratory and Department of Urology, Hospital Clínic, Institut d'Investigacions Biomèdiques August Pi i Sunyer (IDIBAPS), Universitat de Barcelona, 08036 Barcelona, Spain. [112]Centre de Recerca Biomèdica CELLEX, 08036 Barcelona, Spain. [113]Department and Laboratory of Urology, Hospital Clínic, Institut d'Investigacions Biomèdiques August Pi i Sunyer (IDIBAPS), Universitat de Barcelona, 08036 Barcelona, Spain. [114]Genitourinary Program, Moffitt Cancer Center, Tampa, FL 33612, USA. [115]Department of Urology, Klinikum rechts der Isar der Technischen Universitaet Muenchen, 81675 Munich, Germany. [116]Department of General and Clinical Pathology, Alexandrovska University Hospital, Medical University, 1431 Sofia, Bulgaria. [117]Center of Excellence in Genomic Medicine Research, King Abdulaziz University, Jeddah 2252 3270, Saudi Arabia. [118]Grupo de Medicina Xenómica, CIBERER, CIMUS, Universidad de Santiago de Compostela, Avenida de Barcelona, 15782 Santiago de Compostela, Spain. [119]Hérault Cancer Registry, Montpellier cedex 5, 34298 Montpellier, France. [120]Urology Department, Clinique Beau Soleil, 34070 Montpellier, France. [121]INSERM U1147, 75013 Paris, France. [122]Department of Clinical Science, Intervention and Technology, Karolinska Institutet, SE-171 77 Stockholm, Sweden. [123]Swedish Agency for Health Technology Assessment and Assessment of Social Services, SE-102 33 Stockholm, Sweden. [124]Department of Surgical and Perioperative Sciences, Urology and Andrology, Umeå University, SE-901 85 Umeå, Sweden. [125]Department of Surgical Sciences, Uppsala University, SE-751 85 Uppsala, Sweden. [126]Department of Urology, Faculty of Medicine and Health, Örebro University, SE-701 82 Örebro, Sweden. [127]Medical Research Council (MRC) Integrative Epidemiology Unit, University of Bristol, Bristol BS8 2BN, UK. [128]National Institute for Health Research (NIHR) Biomedical Research Centre, University of Bristol, Bristol BS8 1TH, UK. [129]Department of Urology, Cancer Therapy and Research Center, University of Texas Health Science Center at San Antonio, San Antonio, TX 78229, USA. [130]Menzies Health Institute Queensland, Griffith University, Gold Coast, QLD 4222, Australia. [131]Cancer Council Queensland, Fortitude Valley, QLD 4006, Australia. [132]Chris O'Brien Lifehouse (COBLH), Camperdown, Sydney, NSW 2010, Australia. [133]Garvan Institute of Medical Research, Sydney, NSW 2010, Australia. [134]Dame Roma Mitchell Cancer Research Centre, University of Adelaide, Adelaide, SA 5005, Australia. [135]Department of Anatomy and Developmental Biology, Biomedicine Discovery Institute, Monash University, Melbourne, VIC 3800, Australia. [136]Prostate Cancer Translational Research Program, Cancer Research Division, Peter MacCallum Cancer Centre, Melbourne, VIC 3000, Australia. [137]Department of Molecular Medicine and Surgery, Karolinska Institutet, and Department of Urology, Karolinska University Hospital, 171 76 Stockholm, Sweden. [138]Department of Clinical Sciences at Danderyd Hospital, Karolinska Institutet, 182 88 Stockholm, Sweden. [139]Cancer Genome Project, Wellcome Trust Sanger Institute, Hinxton, Cambridge CB10 1SA, UK. [140]Department of Urology, Tampere University Hospital, University of Tampere, Kalevantie 4, FI-33014 Tampere, Finland. [141]Faculty of Medicine and Life Sciences, University of Tampere, FI-33014 Tampere, Finland. [142]Department of Epidemiology, School of Health Sciences, University of Tampere, FI-33014 Tampere, Finland. [143]University of Cambridge Department of Oncology, Oncology Centre, Cambridge University Hospitals NHS Foundation Trust, Cambridge CB1 8RN, UK. [144]Washington University School of Medicine, StLouis, MO 63110, USA. [145]Department of Urology, Aarhus University Hospital, 8200 Aarhus N, Denmark. [146]Department of Radiation Oncology, University of Rochester Medical Center, Rochester, NY 14620, USA. [147]Department of Pathology and Pediatrics, Albert Einstein College of Medicine, Bronx, NY 10461, USA. [148]Second Military Medical University, 200433 Shanghai, China. [149]Wuxi Second Hospital, Nanjing Medical University, 214003 Wuxi, Jiangzhu, China. [150]Department of Urology, The First Affiliated Hospital, Chongqing Medical University, 200032

Chongqing, China. [151]The People's Hospital of Liaoning Province and The People's Hospital of China Medical University, 110001 Shenyang, China. [152]Department of Urology, Zhongshan Hospital, Fudan University Medical College, 200032 Shanghai, China. [153]Precision Medicine, School and Clinical Sciences at Monash Health, Monash University, Clayton, VIC 3168, Australia. [154]Menzies Institute for Medical Research, University of Tasmania, Hobart, TAS 7000, Australia. [155]Faculty of Medical Science, John Radcliffe Hospital, University of Oxford, Oxford OX1 2JD, UK. [156]Department of Radiation Oncology, Complexo Hospitalario Universitario de Santiago, SERGAS, 15706 Santiago de Compostela, Spain. [157]Division of Family Medicine, Department of Neurobiology, Care Science and Society, Karolinska Institutet, Huddinge, SE-171 77 Stockholm, Sweden. [158]Scandinavian Development Services, 182 33 Danderyd, Sweden. [159]University of Cantabria-IDIVAL, 39005 Santander, Spain. [160]School of Public Health, Louisiana State University Health Sciences Center, New Orleans, LA 70112, USA. [161]National Human Genome Research Institute, National Institutes of Health, Bethesda, MD 20892, USA. [162]Department of Urology, Herlev and Gentofte Hospital, Copenhagen University Hospital, Herlev, 2200 Copenhagen, Denmark. [163]Copenhagen Prostate Cancer Center, Department of Urology, Rigshospitalet, Copenhagen University Hospital, DK-2730 Herlev, Denmark. [164]Saarland Cancer Registry, 66119 Saarbrücken, Germany. [165]Department of Urology, University Hospital Ulm, 89075 Ulm, Germany. [166]Cancer Prevention Institute of California, Fremont, CA 94538, USA. [167]Department of Health Research and Policy (Epidemiology) and Stanford Cancer Institute, Stanford University School of Medicine, Stanford, CA 94305-5101, USA. [168]Department of Radiotherapy, Ghent University Hospital, B-9000 Gent, Belgium. [169]Cancer Research Malaysia (CRM), Outpatient Centre, Subang Jaya Medical Centre, 47500 Subang Jaya, Selangor, Malaysia. [170]Urogenital Unit, Division of Medical Oncology, Department of Oncology, University Hospital Centre Zagreb, Šalata 2, 10000 Zagreb, Croatia. [171]Department of Urology, University Hospital Center Zagreb, University of Zagreb School of Medicine, Šalata 2, 10000 Zagreb, Croatia. [172]Department of Urology and Alexandrovska University Hospital, Medical University of Sofia, 1431 Sofia, Bulgaria. [173]Department of Urology, University Hospitals Leuven, BE-3000 Leuven, Belgium. [174]Southampton General Hospital, The University of Southampton, Southampton SO16 6YD, UK. [175]Genetic Oncology Unit, CHUVI Hospital, Instituto de Investigación Biomédica Galicia Sur (IISGS), Complexo Hospitalario Universitario de Vigo, 36204 Vigo (Pontevedra), Spain. [176]Moores Cancer Center, Department of Family Medicine and Public Health, University of California San Diego, La Jolla, CA 92093-0012, USA. [177]Department of Clinical Chemistry, Erasmus University Medical Center, 3015 CE Rotterdam, The Netherlands. [178]Program for Personalized Cancer Care, NorthShore University HealthSystem, Evanston, IL 60201, USA. [179]Department of Epidemiology, University of Washington, Seattle, WA 98195, USA. [180]Department of Epidemiology and Biostatistics, School of Public Health, Imperial College, London SW7 2AZ, UK. [181]Genomics England, Queen Mary University of London, Dawson Hall, Charterhouse Square, London EC1M 6BQ, UK. [182]Genomic Epidemiology Group, German Cancer Research Center (DKFZ), D-69120 Heidelberg, Germany. [183]Epidemiology Program, University of Hawaii Cancer Center, Honolulu, HI 96813, USA. [184]Dana-Farber Cancer Institute, Boston, MA 02215, USA. [185]Paris-Sud University, UMRS 1018, Cedex 94807 Villejuif, France. Deceased: Brian E. Henderson.

