## [Peer Review File · Nature Communications]

Reviewers' Comments:

Reviewer #1:

Remarks to the Author:

This is generally a very well written and extensive analysis of an important component of risk for prostate cancer. The approach seems logical and well conducted. I have some minor and specific comments.

1. the comment lines 396 forward 'For regions which have been fine-mapped using OncoArray meta-analysis data, we used the updated representative lead variants, otherwise the originally reported variant was included provided that it had replicated at genome-wide significance in the meta-analysis; this identified a total of 175 independently associated PCa variants for the FRR calculation (Schumacher et al, in review Nature Genetics, 2017; Dadaev et al., in review Nat Comm, 2017).' is very vague and would be hard to reconstruct. Could you present a supplementary table showing exactly which SNPs were retained and refer to that table.

Figure 1. The graph is low quality and the resolution needs to be improved. I also found the number of meaningless horizontal dotted lines in the first two panels distracting and there was a lack of detail about what the last panel means so I could not interpret it from the legend. Please add more information about it.

I found it very surprising that haplotype analyses were conducted but no results other than a passing remark about one snp being on a specific haplotype block was made. Please provide in a supplementary table or a main table results for haplotypes as these may also shed some light on the risks associated with prostate cancer in this region. I expect sample sizes are smaller and the analysis might be restricted to just Oncoarray data to make the presentation and analysis easier but I think this would provide some insights and also identify some particularly higher risk individuals or perhaps there is some complementarity in risk, but it would be useful to see what happens at the haplotype level.

Reviewer #2:

Remarks to the Author:

The authors fine-mapped the 8q24 susceptibility region in 71,535 prostate cancer cases and 52,935 controls of European ancestry to define the contribution of the 8q24 region to familial risk of prostate cancer and to capture the 8q24 locus-conferred risk as a polygenic risk score. They report a significant increased prostate cancer risk (about 4-fold) among men with a top 1% risk score, and they also estimate that the identified 12 independent risk variants in 8q24 can explain about 11.5% of the familial prostate cancer risk or 25% of this risk related to known genetics. The manuscript is concise and well-written.

As the authors point out there have been multiple previous publications that assessed the contribution of the 8q24 locus to prostate cancer, including estimates for population attributable risk. Combined these studies established that 8q24 is the major risk locus for prostate cancer in diverse populations. The current study analyzed existing data from two large GWAS consortia and generated observations that are an incremental advance to the previous knowledge about 8q24 and prostate cancer.

What remains unclear to the reviewer is how 8q24 improves prediction of prostate cancer risk beyond the knowledge of having familial cancer risk yes/no. Would we expect a high polygenic risk score (top 1%) in men without familial cancer risk? Also, how much of the prostate cancer risk in men of European ancestry is explained by 8q24?

Further comments:

1/ authors performed fine-mapping in 1.4 MB region in 8q24 from 127.6-129.0. The reviewer wonders why such a window has been selected. An important gene, FAM84B, which was discussed in the paper because of the association signal with rs1914295, is not considered when focusing on this region. Is the selection due to the insufficient coverage with the OncoArray? I would suggest extending the region a little bit further to include FAM84B, providing more information with

regards to rs1914295.

2/ in Schumacher et al, currently in review by Nature Genetics, there are more cohorts including UK stage1/2, CaPS1/2, BPC3 and NCI-PEGASUS, why did the authors exclude those cohorts when conducting meta-analysis?

3/ the imputation strategy needs to be carefully evaluated for MAF with 0.1%. as I understand the study and the results are mainly focused on SNPs with at least 1%, then the imputation with 0.1% seems to be a waste of computation time. What is the error rate with the current genotyping platform? Can it be accuracy enough to impute rare event?

4/ author reported 1268 SNPs with p-value 10^{-8} and 2772 SNPs with p-value 0.05, please break down the signals by MAF, to check how many of the signals are coming from the SNPs less than 1%, were they located in functional regions? Could we apply tests to those rare variants, such as the Burden test?

5/ in the statistical analysis section, authors mentioned a modified forward and backward stepwise model for the selection of 2772 SNPs. Which set of SNPs were first included when performing forward selection? And please also describe how to exclude signals from the backward stepwise model. Which software or package was used for this analysis?

6/ author used JAM to confirm the stepwise results, and identified 50 variants. What is the PRS for these 50 SNPs, does it outperform the 12 described signals?

7/ author mentioned "Cis-gene regulation was 382 evaluated for genes within a 1Mb window using 359 prostate adenoma cases from The Cancer Genome 383 Atlas (TCGA PRAD; <https://gdc-portal.nci.nih.gov>)", author should provide more information about how this analysis has been done.

8/ which program was used to plot Figure 1? Should there be a reference for LocusExplorer? Fig1 does not give clear information about functional annotation, please separate the 8q24 region by blocks or signals and provide as supplementary figures, especially for the novel signals that have been identified by current study.

Reviewer #3:

Remarks to the Author:

This is a very well-written paper presenting results from a deeper investigation of the impact of 8q24 variants on prostate cancer risk. In particular, the authors used data from previously reported projects (OncoArray and iCOGS) to more fully investigate the 8q24 region.

Results section:

The authors state that

"These 5,600 markers capture, at $r^2 \geq 0.8$, 90% and 97% of all variants at 8q24 (127.6-129.0 Mb) with $MAF \geq 1\%$ and $\geq 5\%$, respectively (based on 1KGP Phase 3 EUR panel)."

This should clarify up front that the coverage only refers to European populations. How well do they cover the AFR panel?

The authors also state that

"Of these 12 stepwise signals, three had alleles with extreme risk allele frequencies (RAFs) that conveyed large effects (rs77541621, $RAF=2\%$, $OR=1.85$; rs183373024, $RAF=1\%$, $OR=2.67$; rs190257175, $RAF=99\%$, $OR=1.60$)."

In light of these large effects, it would be good to include the 95% CIs as well. In addition, is it possible to replicate these findings to see if the large ORs are due in part to winner's curse? How much were they reduced by the bias correction?

The following observation of no marginal association but conditional associations for two of the SNPs is a little puzzling.

"For two variants, rs78511380 and rs190257175, the marginal associations were not genome-wide significant and substantially weaker than those in the conditional model. For rs78511380, the

marginal OR was slightly protective (OR=0.97; P=0.027), but reversed direction and was highly statistically significant when conditioning on the other 11 variants (OR=1.19; P=3.5x10⁻¹⁸ 229 ; Table 1)."

The authors suggest that the flipped OR for rs78511380 may reflect its lying on a haplotype. And that rs190257175 does not replicate in the JAM analysis, which seems problematic since this is one of the three 'novel' variants highlighted in the paper. Another possible explanation is simply collinearity, whereby these two significant conditional results from stepwise regression are simply due to an over parameterized model of highly correlated variables.

The use of 'confirm' in the following seems a bit strong, as it suggests replication. But it's really just modeling the same data with a different model.

"To confirm the stepwise results and identify candidate variants for potential functional follow-up..."

The polygenic risk score (PRS) analysis is helpful for understanding the overall potential impact of the 8q24 variants on prostate cancer risk. However, the estimates of overall PRS association may be overestimated because the weights are applied to the same data from which they were generated (i.e., 'in-sample'). The authors mention using a bias correction for the newly discovered variants, although it is unclear whether they just mean for the three novel findings reported here. One could use cross-validation to determine more accurate PRS associations across all of the variants.

For the familial relative risk part, did the authors also evaluate how much variation in prostate cancer is explained by the 8q24 variants? Outside of a few cancer genetic epidemiology groups, it is much more standard in the literature to evaluate and present the genetic variation explained-not the familial relative risk-which may give different results.

Reviewers' comments:

Reviewer #1 (Remarks to the Author):

This is generally a very well written and extensive analysis of an important component of risk for prostate cancer. The approach seems logical and well conducted. I have some minor and specific comments.

1. the comment lines 396 forward 'For regions which have been fine-mapped using OncoArray meta-analysis data, we used the updated representative lead variants, otherwise the originally reported variant was included provided that it had replicated at genome-wide significance in the meta-analysis; this identified a total of 175 independently associated PCa variants for the FRR calculation (Schumacher et al, in review Nature Genetics, 2017; Dadaev et al., in review Nat Comm, 2017).' is very vague and would be hard to reconstruct. Could you present a supplementary table showing exactly which SNPs were retained and refer to that table.

As requested, we now provide a list of these SNPs with rs numbers, position, percentage of FRR explained and source study in Supplementary Table 5.

Figure 1. The graph is low quality and the resolution needs to be improved. I also found the number of meaningless horizontal dotted lines in the first two panels distracting and there was a lack of detail about what the last panel means so I could not interpret it from the legend. Please add more information about it.

In the previous submission we provided Figure 1 within a .docx document, which could perhaps explain the low image quality and resolution experienced by the Reviewer. We have resubmitted a modified figure as a vectorised .pdf file with sharp details and no loss of resolution even at extreme magnification.

The horizontal lines denote the P-values of the variants on the Manhattan plot, and help to highlight that after conditional analysis, all marginally associated variants are captured by the final 12 SNPs identified through stepwise selection at the genome-wide significant threshold ($p < 5 \times 10^{-8}$). We agree with the Reviewer that the density of these lines is too high and distracts from the other information shown on the plot, and therefore have reduced the number of lines shown as they suggest.

We have modified the legend for Figure 1 to provide greater details of the information displayed in each panel. We have also modified the panel nomenclature within the figure itself for greater clarity.

Figure 1. LocusExplorer plots of the 12 variants at 8q24 significantly associated with PCa risk. 'Marginal' and 'Conditional' Manhattan plot panels show marginal and conditional association results respectively. Variant positions (**x-axis**) and $-\log_{10} p$ -values (**y-axis**) are shown, with the red line indicating the threshold for genome-wide significant association with PCa risk ($p \leq 5 \times 10^{-8}$) and blue peaks local estimates of recombination rates. The position of the 12 independent variants is labeled in each plot. Clusters of correlated variants for each independent signal are distinguished using different colors and also depicted on the "LD r^2 Hits" track. Stronger shading indicates greater correlation with the lead variant, with variants not correlated at $r^2 \geq 0.2$ with any lead variant uncolored. Pairwise correlations are based on the European ancestry (EUR) panel from the 1000 Genomes Project (1KGP) Phase 3. The relative position of RefSeq genes and biological annotations are shown in the 'Genes' and 'Biofeatures'

panels respectively. Genes on the positive strand are denoted in green and those on the negative strand in purple. Annotations displayed are: histone modifications in ENCODE tier 1 cell lines (Histone track), the positions of any variants that were eQTLs with prostate tumor expression in TCGA prostate adenocarcinoma samples and the respective genes for which expression is altered (eQTL track), chromatin state categorizations in the PrEC cell-line by ChromHMM (ChromHMM track), the position of conserved element peaks (Conserved track) and the position of DNaseI hypersensitivity site peaks in ENCODE prostate cell-lines (DNaseI track). The data displayed in this plot may be explored interactively through the LocusExplorer application (<http://www.oncogenetics.icr.ac.uk/8q24/>).

I found it very surprising that haplotype analyses were conducted but no results other than a passing remark about one snp being on a specific haplotype block was made. Please provide in a supplementary table or a main table results for haplotypes as these may also shed some light on the risks associated with prostate cancer in this region. I expect sample sizes are smaller and the analysis might be restricted to just Oncoarray data to make the presentation and analysis easier but I think this would provide some insights and also identify some particularly higher risk individuals or perhaps there is some complementarity in risk, but it would be useful to see what happens at the haplotype level.

We have provided the results of the haplotype analysis in Supplementary Table 1. As captured by the PRS analysis, men carrying haplotypes with multiple risk alleles are at much higher risk of PCa. However, as shown in this table, the estimated risk does not reflect any additional risk due to alleles occurring on the same haplotype. This has been explained in the Results at the end of page 6:

“The haplotype analysis showed an additive effect of the 12 independent risk variants consistent with that predicted in the single variant analysis; co-occurrence of the 8q24 risk alleles on the same haplotype does not further increase the risk of PCa (Supplementary Table 1). The unique haplotype in block 2 carrying the reference allele for rs190257175 (GCTTAT, 0.5% frequency) is also the sole haplotype associated with a reduced risk of PCa, suggesting that having the C allele confers a protective effect.”

Reviewer #2 (Remarks to the Author):

The authors fine-mapped the 8q24 susceptibility region in 71,535 prostate cancer cases and 52,935 controls of European ancestry to define the contribution of the 8q24 region to familial risk of prostate cancer and to capture the 8q24 locus-conferred risk as a polygenic risk score. They report a significant increased prostate cancer risk (about 4-fold) among men with a top 1% risk score, and they also estimate that the identified 12 independent risk variants in 8q24 can explain about 11.5% of the familial prostate cancer risk or 25% of this risk related to known genetics. The manuscript is concise and well-written. As the authors point out there have been multiple previous publications that assessed the contribution of the 8q24 locus to prostate cancer, including estimates for population attributable risk. Combined these studies established that 8q24 is the major risk locus for prostate cancer in diverse populations. The current study analyzed existing data from two large GWAS consortia and generated observations that are an incremental advance to the previous knowledge about 8q24 and prostate cancer.

What remains unclear to the reviewer is how 8q24 improves prediction of prostate cancer risk beyond the knowledge of having familial cancer risk yes/no.

Although association tests are important for marker discovery, they are not the most appropriate measures for evaluating the predictive value of genetic profiles. We performed an AUC receiver operating characteristic (ROC) analysis using 56,495 samples (33,711 cases and 22,784 controls) from the OncoArray dataset with available information on family history of prostate cancer (yes/no). We compared different scenarios (family history vs. family history + 12 SNPs) to understand how the independent 8q24 risk variants identified in this study improve prediction of prostate cancer beyond having a family history of the disease. The prediction improves substantially when adding the 12 risk variants to the model compared with the model with only family history of prostate cancer (AUC=0.62 vs 0.55). Family history of prostate cancer is known to be an important predictor of an individual’s risk; however, these results show that the prediction is further enhanced when risk variants from 8q24 are also incorporated into the model, and therefore that both are important predictors of prostate cancer risk and should be combined in future risk prediction methodologies. However, since our primary goal is to use the entire dataset for refining the number of independent hits at 8q24 and there are issues in evaluating prediction in the same sample used for model development, we have decided not to include this in the paper. We will evaluate prediction for all variants in an independent data set in the future once additional samples have been collected and genotyped.

Would we expect a high polygenic risk score (top 1%) in men without familial cancer risk?

We would logically expect a degree of correlation between higher polygenic risk score (PRS) and greater likelihood of family history of prostate cancer; however, we would caution that there are a number of important caveats in a polygenic disease such as prostate cancer that could influence this outcome, and thus render the observation of a high PRS for these 12 8q24 risk variants in an individual with unknown family history an entirely plausible outcome. The binary and self-reported family history data available for this analysis may also be insufficient to enable detailed examination of the relationship between PRS and familial risk.

Using available data for first degree family history of prostate cancer in cases and controls from the OncoArray dataset (46183=no, 10312=yes), we performed PRS analysis using the 12 risk variants stratified by family history (yes/no). As shown in the tables below, the RRs for men in the top 1% of the PRS are only slightly greater among those with family history compared with those without family history (FamHist=yes: OR=4.24, 95%CI=2.85-6.31, $p=1.20 \times 10^{-12}$; FamHist=no: OR=3.38, 95%CI=2.88-3.97, $P=3.04 \times 10^{-50}$). This may suggest that PRS is not predictive for family history of prostate cancer; however, one must be cautious of the self-reported family history data as prostate cancer is late onset and the disease may not yet have occurred in family members of the affected individuals.

To clarify this in the paper, we now state: “Risk estimates by PRS category are not modified by family history (FamHist=yes: OR=4.24, 95%CI=2.85-6.31; FamHist=no: OR=3.38, 95%CI=2.88-3.97, data not shown).”

Men with FH=0 (n=46,183)

Risk category percentile	controls	cases	OR (95%CI)	P-values
----------	-------	------------	----------

<1%	201	152	0.63 (0.51; 0.79)	4.53E-05
1%-10%	1800	1369	0.63 (0.59; 0.68)	5.48E-31
10%-25%	3001	2712	0.76 (0.72; 0.81)	3.24E-19
25%-75%	10001	12023	1 (Ref)	
75%-90%	3001	5004	1.39 (1.31; 1.46)	1.04E-32
90%-99%	1799	4096	1.88 (1.76; 2.00)	2.82E-85
>99%	201	823	3.38 (2.88; 3.97)	3.04E-50

Men with FH=1 (n=10,312)

Risk category percentile	controls	cases	OR (95%CI)	P-values
<1%	28	19	0.27 (0.15; 0.49)	2.23E-05
1%-10%	250	443	0.71 (0.60; 0.85)	1.11E-04
10%-25%	417	731	0.73 (0.63; 0.84)	7.59E-06
25%-75%	1393	3397	1 (Ref)	
75%-90%	414	1452	1.40 (1.23; 1.59)	3.89E-07
90%-99%	250	1200	1.94 (1.67; 2.27)	1.44E-17
>99%	28	290	4.24 (2.85; 6.32)	1.20E-12

Also, how much of the prostate cancer risk in men of European ancestry is explained by 8q24?

Based on population attributable risk (PAR) calculations, we estimated that 93.7% (CI=91.2-95.3) of prostate cancer could be explained by the 12 risk alleles at 8q24. However, we would caution that the computed PAR refers to a comparison of individuals that carry none of the risk alleles vs. individuals that carry all of the 12 8q24 risk variants, which is really an uninformative comparison, as very few individuals carry all or none of the risk alleles (see haplotype frequencies in Supp Table 1). In accordance with our finding, other reports of newly identified loci have also emphasized large PARs when comparing extreme genetic profiles (PMID: 18199855, 18565871), and one must be caution in interpreting these results in the context of risk prediction – extreme statistical significance, large relative risks and high PAR do not necessarily assure that a genetic profile will be clinically useful (PMID: 18852206). Establishing that a number of genetic polymorphisms in single or multiple loci are strongly associated with cancer risk through relative risks or PAR estimates is not sufficient. Other parameters estimated in an independent data set, such as sensitivity, specificity, and positive and negative predictive values can be used to estimate the ability of a genetic profile (the 12 risk variants at 8q24, in this case) to discriminate between those who will develop disease and those who will not, as well as their ability to predict individual risk (PMID: 19238176). For example, data from prospective cohort studies should be used to estimate positive and negative predictive values directly, and this is what we plan to do in future studies. Regarding our manuscript, since we know that the PAR calculation does not provide a real estimate of risk prediction for the 12 8q24 SNPs, we do not mention this in the discussion. To gauge the overall contribution of the 8q24 region to genetic risk, please see our response below regarding the proportion of FRR explained and heritability.

Further comments:

1/ authors performed fine-mapping in 1.4 MB region in 8q24 from 127.6-129.0. The reviewer wonders why such a window has been selected. An important gene, FAM84B, which was discussed in the paper because of the association signal with rs1914295, is not considered when focusing on this region. Is the selection due to the insufficient coverage with the OncoArray? I would suggest extending the region a little bit further to include FAM84B, providing more information with regards to rs1914295.

The Reviewer raises a very astute observation and is also correct in their supposition that this constraint is indeed due to insufficient coverage beyond the 127.6-129.0 Mb region that we specified for this analysis. We selected the 127.6-129.0 Mb region for the analysis because both the Oncoarray and iCOGS arrays were designed to include fine-mapping SNPs at 8q24 in this window. These boundaries for dense genotyping on the iCOGS and OncoArray genotyping platforms had however been decided upon based on the results from earlier GWAS studies (of prostate cancer and other cancers, for which 8q24 is a shared risk locus and contributed to the design of the markers on these chips) that have provided no evidence for association with risk beyond these co-ordinates.

We provide below a plot of the variant coverage, extended to also encompass the 0.3 Mb region upstream (127.3-129.0 Mb), which demonstrates the drop in variant coverage in the proximity of FAM84B. Although coverage for common alleles (>5%) is excellent, coverage for less common alleles is lower for this downstream 0.3 Mb region (see plot below). As requested, we have performed an additional analysis on this wider region including FAM84B (for a total of 5,876 SNPs analyzed), which did not alter our results – no additional variant aside from the 12 SNPs we report was independently associated with PCa risk when including this additional region. We therefore conclude that there is relatively low likelihood for additional risk signals physically situated within the 127.3-127.6 Mb region encompassing FAM84B. Accordingly, we prefer to keep the original region boundary encompassing high variant coverage only for the analysis presented within the manuscript.

2/ in Schumacher et al, currently in review by Nature Genetics, there are more cohorts including UK stage1/2, CaPS1/2, BPC3 and NCI-PEGASUS, why did the authors exclude those cohorts when conducting meta-analysis?

The Oncoarray and iCOGS studies were prioritized because the arrays used in these custom genotyping array projects over-selected SNPs at 8q24 for the purpose of fine-mapping, whereas the other studies used standard GWAS arrays comprising sparse tag SNP data, designed to facilitate locus discovery only. This greater level of coverage and precision of genotypes is important when trying to distinguish stronger signals from correlated markers through fine-mapping. These two studies do however comprise 81.6% of the samples in the Schumacher analysis, which we believe represents the most appropriate balance between statistical power and precision from the available sample cohorts from the Schumacher et al. meta-analysis.

3/ the imputation strategy needs to be carefully evaluated for MAF with 0.1%. as I understand the study and the results are mainly focused on SNPs with at least 1%, then the imputation with 0.1% seems to be a waste of computation time. What is the error rate with the current genotyping platform? Can it be accuracy enough to impute rare event?

The 0.1% frequency was used for imputation primarily due to prior knowledge of a rare risk association within 8q24, which had been reported as having a MAF of ~0.5% in EUR controls and a high effect size (rs188140481; PMID: 23104005). As we were imputing a single region, computational burden during imputation was not a primary concern for this project. We therefore used a low MAF cutoff for imputation and relied primarily on imputation quality score to remove low quality variants (of which a greater proportion would inevitably be rare) because we did not want to artificially exclude this important association signal and any other sufficient quality rarer variants that might contribute within the wider 8q24 region. Also, because we had a high density of genotyped markers and haplotypes as a backbone to impute from at 8q24 in the two studies (OncoArray and iCOGS), our prospects to impute lower MAF variants would be maximized, and therefore we would not wish to not consider potentially important variants in these analyses solely due to applying an overly conservative MAF threshold.

The Reviewer is correct though in noting that GWAS are inherently more strongly powered to detect associations with common variants, and therefore that detection of rare variants necessitates their exerting a larger effect size. We estimate however that with our available sample cohort, for a variant with MAF 0.01, we had >95% power to detect an association providing the effect size was OR ≥ 1.5 and assuming a multiplicative model. Whilst variants with MAF <1% are also the most likely MAF class to fail quality control, in addition to these power constraints, we considered that analysis of rare variants passing QC thresholds for imputation quality score (IQS) was appropriate for the 8q24 region based on the prior knowledge of the presence of higher effect size variants in this frequency range within this region. We note that we were able to detect three associations with MAF ≤ 0.02 among the 12 conditionally associated variants we report, and also that we have identified a stronger statistical and functional candidate for the rs188140481 signal – rs183373024 – which was previously known to lie on a 3 SNP haplotype with the original hit [PMID: 27262462] and demonstrates comparable pairwise LD r^2 in our imputed data to 1000 genomes phase 3 EUR samples ($r^2=0.89$). We believe that this justifies our approach; however, we cannot exclude that other rare variants could exist that might also confer differential prostate cancer risk that we either were not able to successfully impute or detect within our sample size due to lower effect size. We have therefore added additional description of this limitation at the end of our Discussion section:

“Our study was predominantly powered to analyze variants with MAF>1% as the imputed variants with MAF=0.1-1% were most likely to fail QC; however, the high density of genotyped markers and haplotypes at 8q24 in the OncoArray and iCOGS studies provided a robust backbone for imputation and increased the chances to impute lower MAF variants with high imputation quality score.”

4/ author reported 1268 SNPs with p-value 10^{-8} and 2772 SNPs with p-value 0.05, please break down the signals by MAF, to check how many of the signals are coming from the SNPs less than 1%, were they located in functional regions? Could we apply tests to those rare variants, such as the Burden test?

Of the 2,772 SNPs significantly associated with PCa risk at $p < 0.05$, 83 have MAF <1%. Of the 1,268 SNPs associated at $p < 10^{-8}$, 2 have MAF <1%. The proportion of nominally significant SNPs in each MAF bin would however be distorted by any signals with a large number of correlated variants even though only one is actually causal, therefore we would be wary of interpreting these numbers directly as relating to numbers of signals coming from each MAF category. We consider the MAFs of the 12 independently

associated lead variants as a more appropriate guide as to the allele frequencies attributable to the risk signals at 8q24. The MAFs of the 12 independent lead variants (shown in Table 1) range from ~1-49%, with 3 variants having $MAF \leq 2\%$ and 8 $MAF > 10\%$; therefore, it appears fairly compelling that PCa risk at 8q24 is likely to be modulated by a combination of common and low frequency/rare variants.

We could in principle apply a burden test for the rare variants within the 8q24 region. We would caution however that with respect to GWAS data, this may be liable to give rise to misleading interpretation. In particular, the individual variant level association analysis demonstrates that two of the association signals – rs77541621 and rs183373024 – are based around rare variants, which would suggest a realistic probability that an aggregated 8q24 rare variant test across the wide interval would also associate with risk, even if no other rare variants beyond these two were in fact associated. Even were these two known rare variants to be excluded from the test, due to LD with other variants, there would also be a high likelihood that the aggregated 8q24 rare variant test would also associate with risk, even if many of those variants are only loosely correlated to the statistically most likely (more common) variants and are highly unlikely to be the causal variant themselves. We also note the likelihood of some degree of correlation between subsets of the rare variants that would be included in the test (unless LD pruning were performed, which may introduce bias), the wide genomic interval that these variants would span, the lack of shared factors between the majority of them to justify their partitioning together aside from their low MAF, and the fact that inclusion of only marginally associated rare variants would inherently bias this test. For these reasons, we would not consider an aggregated rare variant association test prudent for this class of data. We are also not convinced that functional annotation on rare variants would necessarily be informative of anything useful as both uncommon and common variants have been shown to be functional in different GWAS regions (PMID: 27294245).

5/ in the statistical analysis section, authors mentioned a modified forward and backward stepwise model for the selection of 2772 SNPs. Which set of SNPs were first included when performing forward selection? And please also describe how to exclude signals from the backward stepwise model. Which software of package was used for this analysis?

SNPs were included in the stepwise model according to decreasing marginal meta p-value; thus, we first included SNPs with the strongest association. This has been clarified in the footnotes of Table 1: *“Each variant was incorporated in the stepwise model based on the strength of marginal association from the meta-analysis of OncoArray and iCOGS data.”*

We created an R script for the meta stepwise selection based on the following steps. We initially performed 2772 single SNP models each including a candidate SNP plus covariates. We ran the single SNP analysis for each study (OncoArray and iCOGS) separately and then performed a fixed-effect meta-analysis. Then we picked the model with the candidate SNP with the lowest meta p-value and below the genome-wide significant threshold ($P < 10^{-8}$), and repeated the same steps for the two SNP model, three SNP model and so on until the meta p-value of the last included SNP was above the threshold. For backward elimination, if we had one or more fix SNPs with meta p-values no longer statistically significant ($P > 10^{-8}$) after a candidate SNP is added to the model, then we removed the SNP with the least significant p-value among those above the threshold. We applied the same backward elimination principle at each step.

The description on meta stepwise selection in the statistical methods has been improved to mention both the forward and backward elimination procedure:

“A modified forward and backward stepwise model selection with inclusion and exclusion criteria of $p \leq 5 \times 10^{-8}$ was performed on variants marginally associated with PCa risk from the meta results ($p < 0.05$, $n = 2,772$). At each step, the effect estimates for the candidate variants from both studies (OncoArray and iCOGS) were meta-analyzed and each variant was incorporated into the model based on the strength of association. All remaining variants were included one-at-a-time into the logistic regression model conditioning on those already incorporated in the model. We applied a conservative threshold for independent associations, with variants kept in the model if their meta p -value was genome-wide significant at $p \leq 5 \times 10^{-8}$ after adjustment for the other variants in the model.”

6/ author used JAM to confirm the stepwise results, and identified 50 variants. What is the PRS for these 50 SNPs, does it outperform the 12 described signals?

The 50 SNPs that define the JAM credible set are still largely correlated with each other; pruning prior to this analysis was performed at the $r^2 > 0.9$ threshold, therefore many of these SNPs remain moderately or highly correlated to others in the credible set (up to $r^2 = 0.89$ level) and cannot be considered as entirely independent. We would therefore expect the PRS calculated using these 50 SNPs to remain similar to that with the 12 independent signals within this region (used to calculate the score at present) due to attenuation of the effects of the additional included variants (that are partly correlated to the 12 “best” variants) during conditional analysis. Any apparent improvement in the score would more likely reflect a false inflation due to the inclusion of extra variants that aren't independently associated and could therefore be misleading. For this reason, we believe that a PRS analysis on these 50 SNPs would not add any additional information to the study and only report and discuss the original PRS calculated for the 12 independently associated lead variants within the manuscript.

7/ author mentioned “Cis-gene regulation was evaluated for genes within a 1Mb window using 359 prostate adenoma cases from The Cancer Genome Atlas (TCGA PRAD; <https://gdc-portal.nci.nih.gov>)”, author should provide more information about how this analysis has been done.

Detailed information regarding the cis-eQTL analysis is available in Dadaev et al. (PMID:29892050). As this analysis did not majorly influence the findings described in this paper, we preferred to keep this description succinct within our methods and refer interested parties to the companion publication instead. We have added extra text to the methods section to explain more clearly that this information is available within the Dadaev et al. publication; we would however be happy to provide full description of this methodology within this manuscript itself should an editor believe that it would be beneficial for the reader. The eQTL analysis section from the Dadaev et al. paper is provided below for the reviewer's information, and to assist an editor in deciding whether a direction to a citation or direct inclusion of this information would be the preferred way of describing this aspect of our annotation within this paper.

“eQTL analysis

Genotype and gene expression data for 494 samples with PrCa were downloaded from The Cancer Genome Atlas (TCGA; <https://gdc-portal.nci.nih.gov>). For the genotype dataset, quality Control (QC) was performed according to the protocol suggested by Anderson *et al.*⁶⁵, removing samples with heterozygosity >2 standard deviations from the mean, individuals with low genotype call rate (<95%), non-male samples and related or duplicated samples (individuals with identity-by-descent >0.185). Variants with call rate <95% were also excluded from analysis. Principal Component (PC) Analysis was performed to induce the ancestry of the TCGA samples, using the 494 TCGA samples plus 2,504 samples from the 1000 Genomes Project Phase3, with non-European or Finnish samples removed from the analysis. In total, 108 samples and 106 SNPs were removed after performing QC on genotype data. For the expression dataset, we observed that samples from two plates (A31K and A30D) exhibited values substantially higher than samples on the remainder of plates, therefore samples on these plates were also excluded (27 additional samples). Out of the 494 samples, 359 therefore passed QC. Genotypes for samples passing QC were subsequently imputed to the 1000 Genomes Project Phase3 reference panel within the region boundaries applied to the fine-mapping dataset using IMPUTE2. 227,773 variants within the fine-mapping dataset pass QC thresholds in the TCGA imputed data and therefore were available for eQTL analysis. Genes with mean expression across samples of ≤ 6 counts or with expression variance = 0 were also excluded (4,123 and 370 genes removed respectively). Finally, expression values were quantile-normalized by samples and rank-transformed by genes. In total, 16,038 genes passed QC out of the initial 20,531.

For the eQTL analysis, 35 PEER factors⁶⁶ for the top 10,000 expressed genes were used as covariates, plus 3 genotyping PCs. eQTL analysis was performed for each region individually using FastQTL⁶⁷ with 1,000 permutations and a window of 1 megabase from the transcription start site of each gene. Co-localization tests between the eQTLs and GWAS SNPs were then performed following the approach suggested by Nica *et al.*⁶⁸. First, for each significant eQTL, we added the imputed SNP to the linear regression to assess if the inclusion better explains the change in expression of the gene.

expression \sim genotype(eQTL) + cov + genotype(imp. SNP)

We retrieved the *P*-value of this new linear regression, assigning *P*-value of 1 if the eQTL and imputed SNP are the same variant. Secondly, we ranked the *P*-values in descending order for each eQTL. Finally, we calculated the co-localization score for each pair of eQTL and imputed SNPs as:

colocalization score = $(N - \text{rank})/N$

where *N* is the total number of imputed SNPs in that region and rank is the rank of the imputed SNP we are including. In general, if an eQTL and an imputed SNP represent the same signal, this will be reflected by the imputed SNP having a high *P*-value, a low rank, and consequently a high co-localization score.”

8/ which program was used to plot Figure 1? Should there be a reference for LocusExplorer? Fig1 does not give clear information about functional annotation, please separate the 8q24 region by blocks or signals and provide as supplementary figures, especially for the novel signals that have been identified by current study.

Figure 1 was plotted using LocusExplorer. We agree with the Reviewer that it may be helpful to reference this software and have added that information in the legend of Figure 1. The Reviewer correctly notes that when plotting the range of genomic coordinates required to display all conditionally associated variants within the 8q24 region, it is not possible to achieve sufficient magnification to display and interpret detailed functional annotations for each variant. The primary intention of the static plot in Figure 1 is to visualize the relative positions of the conditionally associated variants, their correlation or lack thereof with the other independently associated SNPs, that the 12 conditionally associated SNPs capture all risk signals present in the marginal data, and a basic overview of the genomic context of these signals in relation to the positions of genes and prospective regulatory elements. One of the benefits of LocusExplorer is that a user is able to dynamically explore data through the application, customizing the level of zoom and information displayed, and we believe that this is the most appropriate means to further explore this data and visualize the functional context in greater detail. We have made the data underlying this plot available through the LocusExplorer application (<http://www.oncogenetics.icr.ac.uk/8q24/>), and have added this information and the URL to the figure legend to enable readers to further explore our data beyond the level that we can feasibly display within the paper in static figures.

To improve the clarity of Figure 1 we have cut unnecessary regions upstream and downstream of the 12 8q24 hits which allows one to zoom in on the region and reach a clearer visualization of functional annotations overlapping each signal. We have also modified the panel nomenclature within the figure itself for greater clarity, and provided more details in the figure legend about the annotations displayed in the 'Biofeatures' plot.

We also agree with the reviewer's suggestion to further break down the image in Figure 1 by signal as additional supplementary figures encompassing smaller windows of genomic coordinates, and have added this as Supplementary Figure 1.

Reviewer #3 (Remarks to the Author):

This is a very well-written paper presenting results from a deeper investigation of the impact of 8q24 variants on prostate cancer risk. In particular, the authors used data from previously reported projects (OncoArray and iCOGS) to more fully investigate the 8q24 region.

Results section:

The authors state that

"These 5,600 markers capture, at $r^2 > 0.8$, 90% and 97% of all variants at 8q24 (127.6-129.0 Mb) with $MAF \geq 1\%$ and $\geq 5\%$, respectively (based on 1KGP Phase 3 EUR panel)."

This should clarify up front that the coverage only refers to European populations. How well do they cover the AFR panel?

As requested, the 5,600 markers tag, at $r^2 > 0.8$, 3498 (70.7%) out of the 4,945 variants at 8q24 (127.6-129.0 Mb) with $MAF \geq 5\%$ based on 1KGP Phase 3 AFR panel. However, this information is not included in the manuscript as this analysis does not include men of African ancestry.

The authors also state that

"Of these 12 stepwise signals, three had alleles with extreme risk allele frequencies (RAFs) that 208 conveyed large effects (rs77541621, RAF=2%, OR=1.85; rs183373024, RAF=1%, OR=2.67; rs190257175, 209 RAF=99%, OR=1.60)."

In light of these large effects, it would be good to include the 95% CIs as well.

As requested, we added the confidence intervals to the text (which are also provided in Table 1).

In addition, is it possible to replicate these findings to see if the large ORs are due in part to winner's curse?

Although it may be possible that the "relatively" large ORs observed in our study may be due to winner's curse, we believe that this scenario is unlikely in our context. Since associations between 8q24 variants and prostate cancer risk have been reported previously, I would regard our analysis as really a replication study combined with fine-mapping data, and so less prone to the inflation of effect sizes than the far smaller, less well powered discovery phase GWAS studies, and therefore our estimates should be highly accurate. There is the potential for ORs of the lead SNPs for known signals to increase due to fine-mapping towards more likely causal variants, and we are planning to investigate this in a prospective setting using data from multi-ethnic populations.

How much were they reduced by the bias correction?

To answer the Reviewer's question, we compared the uncorrected estimates (maximum likelihood estimates) with the biased corrected estimates for the FRR calculation (see graph below). Conditional estimates were used for the FRR calculation by fitting a single model with all variants from OncoArray data. As the graph shows, the actual bias in the effects estimates is very small due to the extremely large sample size combined with the corresponding small effect sizes for the discovered SNPs.

The following observation of no marginal association but conditional associations for two of the SNPs is a little puzzling.

"For two variants, rs78511380 and rs190257175, the marginal associations were not genome-wide significant and substantially weaker than those in the conditional model. For rs78511380, the marginal OR was slightly protective (OR=0.97; P=0.027), but reversed direction and was highly statistically significant when conditioning on the other 11 variants (OR=1.19; P=3.5x10⁻¹⁸ 229 ; Table 1)."

The authors suggest that the flipped OR for rs78511380 may reflect its lying on a haplotype. And that rs190257175 does not replicate in the JAM analysis, which seems problematic since this is one of the three 'novel' variants highlighted in the paper. Another possible explanation is simply collinearity, whereby these two significant conditional results from stepwise regression are simply due to an over parameterized model of highly correlated variables.

rs190257175 was selected in the 95% credible set by JAM in 2 of the 4 seeds, but with extremely low posterior probability. When filtering the final credible set to remove noise, we used an inclusion criterion of selection in ≥ 3 out of 4 seeds to remove low quality variants. So, rs190257175 wasn't entirely refuted by JAM, although the evidence didn't meet the required levels. rs190257175 is not strongly correlated with any of the 12 independent hits from the final stepwise model ($r^2 < 0.031$).

Similarly, rs78511380 is only weakly correlated with one other variant (rs72725879, $r^2=0.278$) among the 12 independent hits from the stepwise model, with correlation with the remaining variants at $r^2 \leq 0.032$. Based on that, we believe that collinearity is not a plausible explanation for their significant conditional result from stepwise regression.

The marginal association of rs78511380 provides a crude estimate which is not independent of LD or haplotype structure between nearby SNPs. When we look at the haplotype structure of the 12 independent hits at 8q24 (Supplementary Table 1), we notice that the reference allele for rs78511380 (A allele) occurs on a haplotype with risk alleles for rs190257175 (T), rs72725879 (T) and rs5013678 (T) [haplotype GTTTAA, 8%], with all other haplotypes with frequency $\geq 1\%$ carrying the risk allele for rs78511380 (T). As the haplotype GTTTAA is associated with increased risk of prostate cancer (OR= 1.21, 95%CI=1.16-1.26) and is the only haplotype with a substantial frequency carrying the A allele for rs78511380, this hides the positive association with the risk allele (T allele) in the marginal model. This is also demonstrated by comparing the effect size for haplotype GTTTAA with haplotype GTTTAT, which only differ from each other by the presence of the reference/risk allele for rs78511380 (A/T). As expected, the haplotype with the risk allele for rs78511380 (GTTTAT) is associated with a higher risk (OR=1.45, 95%CI=1.39-1.50) compared with the haplotype with the reference A allele (GTTTAA, OR= 1.21, 95%CI=1.16-1.26).

We have provided a better explanation in the text: *“The reference allele for rs78511380 (A, 8% frequency) occurs on a haplotype in block 2 together with the risk alleles for rs190257175, rs72725879 and rs5013678 (haplotype GTTTAA, 8%) which obscures the positive association with the T allele of rs78511380. Thus, the marginal protective effect associated with the risk allele for rs78511380 reflects an increased risk associated with the occurrence on a risk haplotype with other risk alleles (Supplementary Table 1).”*

The use of 'confirm' in the following seems a bit strong, as it suggests replication. But it's really just modeling the same data with a different model.

"To confirm the stepwise results and identify candidate variants for potential functional follow-up..."

We agree with the Reviewer and we have changed the wording as follow: *“We explored our data with a second fine-mapping approach, JAM (Joint Analysis of Marginal summary statistics)¹¹, which uses GWAS summary statistics to identify credible sets of variants that define the independent association signals in susceptibility regions (see Methods).”*

The polygenic risk score (PRS) analysis is helpful for understanding the overall potential impact of the 8q24 variants on prostate cancer risk. However, the estimates of overall PRS association may be overestimated because the weights are applied to the same data from which they were generated (i.e., 'in-sample'). The authors mention using a bias correction for the newly discovered variants, although it is unclear whether they just mean for the three novel findings reported here. One could use cross-validation to determine more accurate PRS associations across all of the variants.

As mentioned in the Methods section, 8q24-only PRS was calculated for variants from the final model (n=12) with allele dosage from OncoArray and iCOGS weighted by the per-allele conditionally adjusted ORs from the meta-analysis – so using conditional effect estimates to account for correlation between SNPs is the real bias correction for all variants in the PRS. Thus, our analysis not only takes advantage of

the fine-mapping data and the large sample size, but also of the conditional estimates that are less biased by LD between SNPs and the inflation of effect size due to the refining of the association signals. We therefore argue that using conditional effect estimates as weights in the PRS analysis is appropriate for estimating the overall impact of the 12 risk variants on prostate cancer risk.

We also agree with the Reviewer that cross-validation would be useful to determine more accurate risk associations and estimate prediction error. Nonetheless, we recognize that estimates of association with the 12 risk variants at 8q24, although independent of other risk variants in the same region, are not sufficient to appropriately evaluate the clinical utility of the 8q24 genotype profile in prostate cancer risk prediction. We argue that the aim of our study was to use GWAS and fine-mapping data for discovery of novel associations rather than risk prediction; thus, using the full dataset is preferable to cross-validation for this study, and we plan to do a cross-validation when other large datasets become available (i.e. UKBiobank).

For the familial relative risk part, did the authors also evaluate how much variation in prostate cancer is explained by the 8q24 variants?

We note that for the calculations of the proportion of FRR we now use the marginal estimates of effect and allele frequency from the overall meta-analysis as reported in the recently published paper by Schumacher et al. (PMID:29892016) – we view these estimates as the most accurate as they use the entire available data for prostate cancer. For regions with multiple SNPs, we continue to use the conditional estimates of effect. We have updated Table 3 to report both the proportion of FRR explained and heritability. The 8q24 variants reported in this study account for 25.4% of what can be currently explained of the familial risk of prostate cancer by known genetic risk factors (37.08%) and they account for 22.2% of the total explained heritability by these known variants (0.118).

Outside of a few cancer genetic epidemiology groups, it is much more standard in the literature to evaluate and present the genetic variation explained-not the familial relative risk-which may give different results.

As mentioned above, we have updated Table 3 to include the heritability of PCa explained by 8q24 and all other risk variants. We have also presented the heritability results together with the FRR results in the text.

Table 3. Proportion of Familial Relative Risk (FRR) and heritability (h_g^2) of PCa explained by known risk variants

Source	No. of variants	Proportion of FRR (95%CI)	% of total FRR	h_g^2 (SE)	% of total h_g^2
8q24 ¹	12	9.42 (8.22-10.88)	25.4	0.027 (0.011)	22.2
HOXB13 ²	1	1.91 (1.20-2.85)	5.2	0.004 (0.005)	3.0
All other variants ^{2,3}	162	25.77 (22.94-29.36)	69.5	0.092 (0.010)	74.9
Total	175	37.08 (32.89-42.49)	100	0.118 (0.012)	100

¹Conditional estimates were derived by fitting a single model with all variants from OncoArray data

²Risk estimates and allele frequencies for regions with a single variant are from a meta-analysis of OncoArray, iCOGS and 6 additional GWAS ³

³Risk variants included from fine-mapping of PCa susceptibility loci in European ancestry populations ¹³

REVIEWERS' COMMENTS:

Reviewer #1 (Remarks to the Author):

I have no further suggestions.

Reviewer #2 (Remarks to the Author):

The authors addressed all questions.

Reviewer #3 conveyed to us in the remarks to the editor that they had no further concerns with this manuscript.